# Phosphate acts directly on the calcium-sensing receptor to stimulate parathyroid hormone secretion

Patricia P. Centeno [1], Amanda Herberger[2], Hee-Chang Mun[3], Chialing Tu[2], Edward F. Nemeth[4], Wenhan Chang[2], Arthur D. Conigrave [3] & Donald T. Ward[1]*

Extracellular phosphate regulates its own renal excretion by eliciting concentration-dependent secretion of parathyroid hormone (PTH). However, the phosphate-sensing mechanism remains unknown and requires elucidation for understanding the aetiology of secondary hyperparathyroidism in chronic kidney disease (CKD). The calcium-sensing receptor (CaSR) is the main controller of PTH secretion and here we show that raising phosphate concentration within the pathophysiologic range for CKD significantly inhibits CaSR activity via non-competitive antagonism. Mutation of residue R62 in anion binding site-1 abolishes phosphate-induced inhibition of CaSR. Further, pathophysiologic phosphate concentrations elicit rapid and reversible increases in PTH secretion from freshly-isolated human parathyroid cells consistent with a receptor-mediated action. The same effect is seen in wild-type murine parathyroid glands, but not in CaSR knockout glands. By sensing moderate changes in extracellular phosphate concentration, the CaSR represents a phosphate sensor in the parathyroid gland, explaining the stimulatory effect of phosphate on PTH secretion.

[1] Faculty of Biology, Medicine and Health, The University of Manchester, Manchester, UK. [2] UCSF Department of Veterans Affairs Medical Center, San Francisco, CA, USA. [3] Charles Perkins Centre, University of Sydney, School of Life and Environmental Sciences, Sydney, NSW, Australia. [4] MetisMedica, 13 Poplar Plains Road, Toronto, ON M4V 2M7, Canada. *email: d.ward@manchester.ac.uk

The homeostatic control of extracellular calcium ($Ca_o^{2+}$) and inorganic phosphate (Pi) concentrations depends primarily on the secretion of parathyroid hormone (PTH). The calcium-sensing receptor (CaSR) represents the key controller of PTH secretion and is expressed abundantly in parathyroid glands and in renal tubules. CaSR maintains $Ca^{2+}$ homeostasis (1.2 mM ionized $Ca^{2+}$) by modulated suppression of PTH secretion and renal $Ca^{2+}$ reabsorption in response to increased $Ca_o^{2+}$[1]. When circulating $Ca_o^{2+}$ levels drop (hypocalcemia), the decrease in parathyroid CaSR activity permits increased PTH secretion which then acts to release $Ca^{2+}$ and Pi from bone[2]. PTH also stimulates Pi excretion in the renal proximal tubule, thus eliminating the released Pi and so permitting ionized $Ca_o^{2+}$ concentration to rise that feeds back on the parathyroid glands to inhibit further PTH secretion[2,3]. In contrast, increased Pi concentration stimulates PTH secretion by a mechanism that remains unclear[4–7].

The phenomenology of Pi-induced stimulation of PTH secretion is well described in vitro and in vivo. It has been reported that Pi elicits concentration-dependent stimulation of PTH from bovine[4] and rat parathyroid tissue[5,6]. In addition, a high-phosphate diet or Pi loading increased serum PTH levels in healthy and in nephrectomized rats[6–8]. However, the molecular mechanism mediating the effect of Pi on PTH secretion remains uncertain and controversial. Pi levels are normally maintained between 0.8 and 1.4 mM by coordinated regulation of intestinal absorption, renal excretion, and influx/efflux from bone. Parathyroid glands and bone can sense increased extracellular Pi, by an unknown mechanism, and respond by secreting PTH and fibroblast growth factor 23 (FGF23) respectively, which then increase renal excretion of Pi[9–14].

The molecular mechanism linking Pi and PTH secretion is relevant for understanding the etiology of secondary hyperparathyroidism (SHPT). SHPT is a common complication of chronic kidney disease (CKD), triggered by hyperphosphatemia, hypocalcemia, and low levels of 1,25OH$_2$D. SHPT is characterized by parathyroid gland hyperplasia that leads to reduced expression of the vitamin D receptor and CaSR, and chronically elevated PTH secretion. In SHPT, chronic underactivation of the CaSR permits continuously elevated levels of PTH secretion causing chronic dysfunction of the $Ca_o^{2+}$ homeostatic system and profound bone loss[15–18]. In association with SHPT, increased Ca × P product contributes to vascular calcification and eventual heart disease, calciphylaxis (tissue necrosis), and renal osteodystrophy[19,20]. Collectively, these various elements of dysfunctional mineral metabolism are referred to as CKD–MBD (mineral bone disorder), which represents one of the most serious complications of renal disease[15,18,21]. In an attempt to reduce CKD mortality and morbidity, national clinical practice guidelines have been developed[22–24].

Currently, the most common therapeutic options for patients with end-stage CKD undergoing dialysis are the calcimimetic drugs cinacalcet or etelcacetide (positive allosteric modulators of the CaSR), phosphate binders, 1,25OH$_2$D supplements, and parathyroidectomy[25]. However, none of these treatments yet provide sufficient amelioration of CKD–MBD to avoid vascular calcification and cardiovascular mortality[19,21,22,25,26].

While the CaSR is the main controller of PTH secretion, its recently crystallized extracellular domain revealed four putative multivalent anion-binding sites occupied by Pi or $SO_4^{2-}$[27]. Of these, sites 1 and 3, based in part on residues R62 and R66, were found exclusively in the inactive conformation, whereas site 4, based partially on residues K225 and R520 was found only in the active conformation. Site 2, based in part on R66 and R69, was observed in both the active and inactive conformations, suggesting a structural role[27]. These observations suggest that anion binding to sites 1 and 3 may preferentially stabilize the inactive conformation of the CaSR. Here we demonstrate that the CaSR represents a phosphate sensor in the parathyroid gland. Specifically, by increasing extracellular Pi, at concentrations observed in CKD, we demonstrate that hyperphosphatemia inhibits the CaSR in a noncompetitive manner and thus increases PTH secretion. These data provide a molecular mechanism for the stimulatory action of high physiological and pathophysiologic Pi levels on PTH secretion.

## Results

**Elevated Pi concentrations inhibit the CaSR.** We first evaluated the effect of acute increases in Pi concentration in CaSR-transfected HEK-293 cells (CaSR-HEK) by measuring CaSR-induced $Ca_i^{2+}$-mobilization upon stimulation with different concentrations of $Ca_o^{2+}$ (Fig. 1a, b). Then, CaSR-specific positive allosteric modulators (PAMs) were included so that the lowest possible $Ca_o^{2+}$ concentration could be used to achieve CaSR-induced $Ca_i^{2+}$ mobilization, thus reducing the risk of Ca × P association in the presence of increased Pi concentrations. Thus, Fura2-loaded CaSR-HEK cells were incubated in experimental buffer containing 0.5 mM $Ca^{2+}$ plus either spermine (Fig. 1c), NPS-R568, hereafter R568, or cinacalcet, a CaSR PAM used clinically (Fig. 1d–f). Cells were switched to buffers containing 0.8 mM Pi (a normal plasma concentration in humans) and then 2 mM Pi (a pathophysiologic concentration observed in CKD patients) before being returned to experimental buffer (without Pi). CaSR responsiveness at 0.8 mM Pi (i.e., 350/380 Fura2 ratio area under the curve) was not significantly decreased when compared with 0 mM Pi (Fig. 1f), whereas the CaSR response in 2 mM Pi was significantly decreased (by 40%) in the experiments performed, regardless of the CaSR agonist or PAM used (Fig. 1c–f). This inhibition was immediately reversible upon removal of Pi in all cases. Furthermore, the decline in CaSR responsiveness elicited by Pi cotreatment was much greater than that for time-matched controls (Fig. 1f). It should be noted that increasing Pi concentration from 0.8 to 2 mM decreased the free ionized $Ca^{2+}$ concentration by 7%, and the Pi inhibitory effect was still observed upon correction of $Ca^{2+}$ levels (Supplementary Fig. 1).

Carbachol induces non-CaSR-mediated $Ca_i^{2+}$-mobilization in CaSR-HEK cells by activating muscarinic acetylcholine receptors (mAChRs) expressed endogenously in the cells. However, Pi cotreatment failed to alter the response to carbachol suggesting that it elicits a specific inhibitory effect on CaSR-mediated signaling (Supplementary Fig. 2a).

If the apparent inhibitory effect of Pi on CaSR was due solely to buffering ionized $Ca^{2+}$, then lowering extracellular pH should reverse this effect by releasing free $Ca^{2+}$. Therefore, $Ca_i^{2+}$-mobilization experiments were repeated but adding a buffer with 2 mM Pi at pH 7.2 (equivalent to moderate-to-severe metabolic acidosis observed in end-stage CKD patients). The results confirmed that even under acidic conditions, increased Pi concentration resulted in a substantial (60%) reduction of CaSR activity that was immediately reversed upon removal of Pi and return to pH 7.4 (Supplementary Fig. 2b). To confirm that the Pi effect was not readout-dependent, we also quantified the effect of Pi on CaSR-induced extracellular signal-regulated kinase phosphorylation (pERK). Indeed, hyperphosphatemia (2 mM Pi) significantly reduced the maximal CaSR-induced pERK in CaSR-HEK cells by 30% and this inhibitory effect was again maintained in acidic conditions (pH 7.2) (Fig. 2a).

**Pi acts as a noncompetitive antagonist of CaSR.** According to the recently reported human CaSR-ECD crystal structure[27], multivalent anions bind to CaSR at defined sites, distinct from the

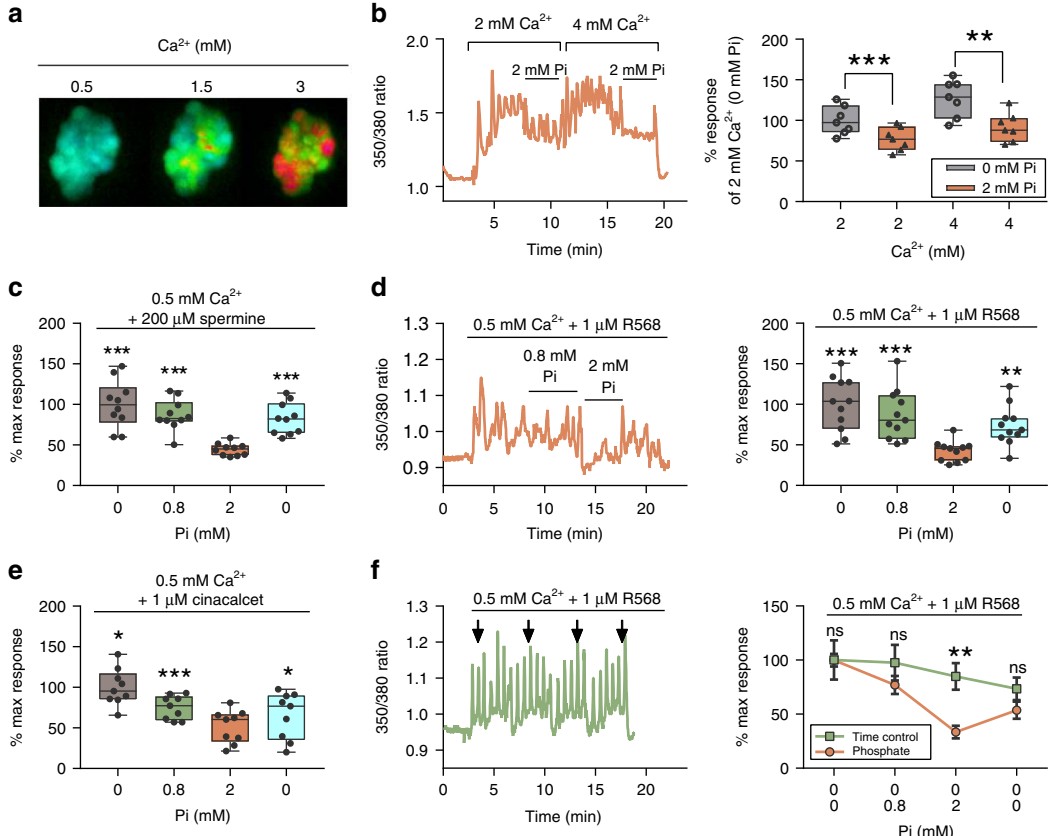

**Fig. 1** Pathophysiologic concentrations of Pi significantly inhibit the CaSR. **a** Representative false-color images from epifluorescence imaging of Fura2-loaded CaSR-HEK cells showing CaSR-mediated $Ca_i^{2+}$ mobilization upon stimulation with increasing $Ca_o^{2+}$ concentrations (as indicated with warm colors). **b–f** Inhibitory effect of Pi on CaSR-mediated $Ca_i^{2+}$ mobilization upon stimulation with different concentrations of $Ca^{2+}$ (**b**), and cotreatment with R568 (**d, f**), spermine (**c**), and cinacalcet (**e**). **b, d, f** (left) Representative $Ca_i^{2+}$ mobilization traces (Fura2 ratio) from a single cell in response to Pi (**b, d**), and time-matched buffer changes without Pi (**f**). **f** Inhibitory effect of Pi on CaSR-mediated $Ca_i^{2+}$ mobilization upon stimulation with R568 compared with time-matched control in the absence of Pi; data shown as %mean ± SEM. Data expressed as percent control of the area under the curve for each treatment, $n = 7$ (**b**), $n = 10$ (**c**), $n = 11$ (**d**), $n = 9$ (**e**), and $n = 9$ (**f**). Data shown in box-and-whisker plots: box ends indicate upper and lower quartiles; midline indicates the median while error bars indicate the range. Data were analyzed by using RM-ANOVA with Dunnett's multiple comparison (**b–e**) or unpaired $t$ test (**f**). ns not significant; *$P < 0.05$, **$P < 0.01$, and ***$P < 0.001$. Source data are provided as a Source Data file

binding sites described for $Ca^{2+}$ or ʟ-amino acids. To elucidate the molecular mechanism underlying Pi-mediated CaSR inhibition, we first examined the $Ca_o^{2+}$ concentration-effect relationship in the absence and presence of 0.8 and 2 mM Pi by measuring $Ca_i^{2+}$-mobilization in CaSR-HEK cells (Fig. 2b). As before, R568 was included to induce CaSR responses at lower $Ca_o^{2+}$ concentrations in order to reduce the risk of Ca × P association. The CaSR maximal response ($E_{max}$) to $Ca_o^{2+}$ was inhibited in the presence of 2 mM Pi (by 32 ± 3%; $n = 8$) and this inhibitory effect was not overcome by further increases in $Ca^{2+}$ concentration. Interestingly, Pi did not alter $Ca_o^{2+}$ potency ($EC_{50}$) (Fig. 2b). Thus, Pi inhibited the CaSR via an apparent noncompetitive antagonism, reducing receptor efficacy but not potency. Moreover, the inhibitory effect of Pi was concentration-dependent following stimulation with either 1 or 1.5 mM $Ca^{2+}$ and 1 μM cinacalcet ($IC_{50}$ 1.21 mM) (Fig. 2c) or with 1.5 mM $Ca^{2+}$ and 1 μM R568 ($IC_{50}$ 1.27 mM) (Supplementary Fig. 3a). These $IC_{50}$ values for Pi indicate CaSR sensitivity to Pi both over its physiological range (0.8–1.4 mM), and also sensitivity to the hyperphosphatemia of CKD, indicating the potential pathophysiological relevance of this effect.

Inorganic $SO_4$ is the other multivalent anion proposed to bind the CaSR ECD according to crystal structure data[27]. $SO_4$ may act

as a ligand at the same sites as Pi, and therefore we tested whether this anion could also inhibit the CaSR. Indeed, $SO_4$ inhibited $Ca_i^{2+}$-mobilization in a concentration-dependent manner with a similar potency to Pi ($IC_{50}$ 1.31 mM) (Fig. 2c). However, as physiological $SO_4$ concentrations are lower than those observed for Pi (~0.3 mM in serum), $SO_4$ would appear to be of lesser physiological significance as a CaSR inhibitor. We also tested different Pi concentrations in the presence of 0.3 mM $SO_4$ (physiological), and the $IC_{50}$ obtained was not significantly different from that obtained with Pi alone (Supplementary Fig. 3b). Interestingly, the inhibitory concentration-effect curves for both Pi and $SO_4$ had Hill slopes close to −1, suggesting the existence of a unique inhibitory anion-binding site, or at least the lack of cooperativity between binding sites (Fig. 2 and Supplementary Fig. 3).

**Pi increases PTH secretion in human parathyroid cells.** Elevated PTH secretion levels together with hyperphosphatemia are major complications of CKD. We measured the direct effect of Pi on PTH secretion from freshly isolated parathyroid cells obtained from samples of healthy human parathyroid tissue. In order to minimize CaSR downregulation, human parathyroid tissue was stored as undigested sliced tissue cubes. Tissue digestion and cell

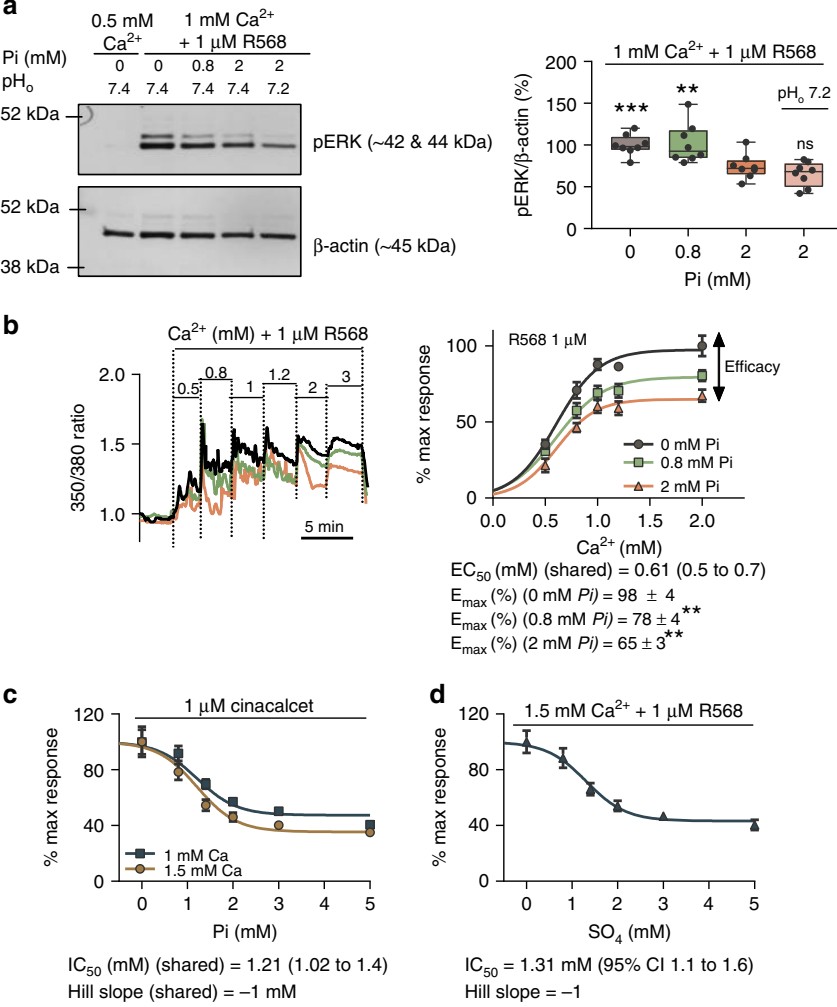

**Fig. 2** Pi inhibits the CaSR in a noncompetitive manner. **a** Representative immunoblot showing Pi-mediated inhibition of CaSR-induced pERK after 10 min of treatment with Pi and acidosis (pH 7.2). Changes in pERK were determined by densitometry, corrected for β-actin abundance and normalized to Pi-free CaSR-stimulated control ($n = 8$ dishes from four experiments; 0.5 mM $Ca^{2+}$ shown as a negative control). **b** Representative traces showing $Ca_i^{2+}$ mobilization from single cells exposed to buffers containing increasing levels of $Ca_o^{2+}$ (left), and $Ca^{2+}$ concentration-effect curves (right) in the presence of 0 ($n = 7$), 0.8 ($n = 9$), and 2 ($n = 10$) mM Pi from three independent experiments. $E_{max}$ expressed as %mean ± SEM. **c** Pi concentration-effect curves for $Ca_i^{2+}$-mobilization upon stimulation with cinacalcet and 1 or 1.5 mM $Ca_o^{2+}$ ($n = 7$, two independent experiments). **d** $SO_4$ concentration-effect curves for $Ca_i^{2+}$-mobilization upon stimulation with R568 and 1.5 mM $Ca_o^{2+}$ ($n = 8$, two independent experiments). **b–d** Area under the curve was calculated for each treatment and normalized to maximal response. Data were fitted to a four-parameter Hill equation (Eq. (1)) for sigmoidal dose response variable slope, and fitted best when $EC_{50}$ (**b**)/$IC_{50}$ (**c, d**), expressed as mean (95% confidence interval), were shared among data sets, $P < 0.01$ extra sum-of-square $F$-test. Data expressed as %mean ± SEM. Data were analyzed by using RM-ANOVA with Dunnett's multiple comparisons, ns not significant; *$P < 0.05$, **$P < 0.01$, and ***$P < 0.001$. Source data are provided as a Source Data file

preparation was performed immediately before PTH assay. In addition, at the beginning of each experiment, we evaluated the responsiveness and quality of the specimen by decreasing $Ca_o^{2+}$ from 1.2 to 1 mM, leading to the expected CaSR-mediated rise in PTH secretion (Supplementary Fig. 4). We measured PTH secretion every 2 min in a perifusion model ($Ca^{2+}$, 1.2 mM) in the presence of increasing Pi concentrations (0.8, 1.4, 2, and 3 mM) encompassing the physiological and pathophysiological range. A significant increase in PTH secretion was observed at 2 mM Pi (hyperphosphatemia), +56 ± 10%, with a further increase at 3 mM Pi (severe hyperphosphatemia), +77 ± 13% (Fig. 3, individual traces shown in Supplementary Fig. 4). This stimulatory effect of Pi on PTH secretion was quickly overcome, within minutes, when Pi concentration was returned to 0.8 mM (physiologic). These rapid and reversible effects are consistent with a receptor-mediated response (Fig. 3). In addition, we

confirmed that the addition of Pi to the buffer had only a marginal effect on ionized $Ca^{2+}$ concentration (−6% 0.8 vs. 2 mM Pi; Supplementary Fig. 1a). We further confirmed the inhibitory effect of Pi on CaSR-induced $Ca_i^{2+}$ mobilization in human parathyroid cells obtained by collagenase digestion of parathyroid tissue (Supplementary Fig. 5).

**Pi increases PTH secretion via the CaSR.** To assess whether the rise in PTH secretion observed in hyperphosphatemia was mediated by the CaSR, we compared PTH secretion levels from control and CaSR-knockout (KO *Casr*) murine parathyroid glands (Fig. 4a). KO *Casr* mice have a homozygous parathyroid gland-specific deletion of exon 7 (transmembrane domain) of the CaSR. To be considered as full KO *Casr*, only mice exhibiting (a) high serum PTH levels (greater than tenfold higher than controls), and (b) lack of a response of their parathyroid glands

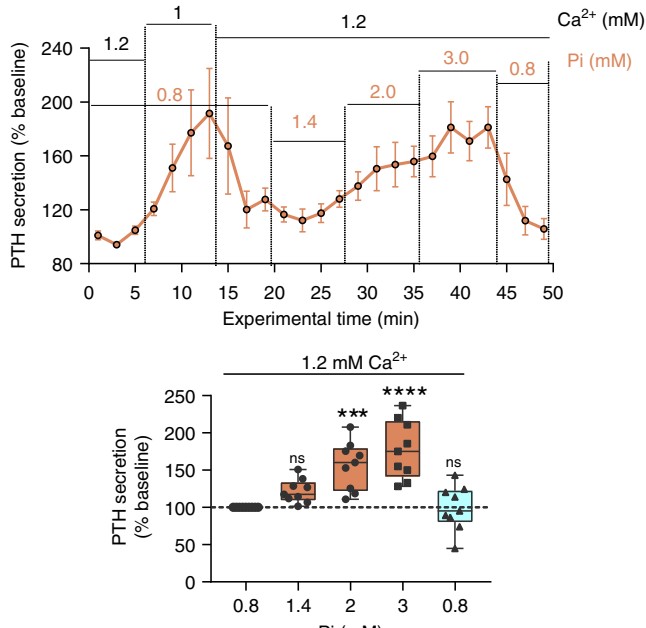

**Fig. 3** Pathophysiologic Pi concentrations increase PTH secretion in human parathyroid cells. Effect of hyperphosphatemia on PTH secretion (measured every 2 min) from perifused, freshly isolated human parathyroid cells (upper). One millimolar $Ca^{2+}$ was used as an internal control to confirm $Ca_o^{2+}$ responsiveness and CaSR expression in the cell preparation. Data normalized to baseline (initial exposure to 1.2 mM $Ca_o^{2+}$/0.8 mM Pi) and shown in box-and-whisker plots. Data from $N = 9$ human samples from nine biologically independent patients; individual traces are shown in Supplementary Fig. 4. ns not significant, [***]$P < 0.001$, [****]$P < 0.0001$ by Friedman's with Dunn's post hoc test. Source data are provided as a Source Data file

to $Ca^{2+}$ concentration changes (<10% response when changing from 1.2 to 0.8 mM or from 0.8 to 1.6 mM) were included in the results (Supplementary Fig. 6)[28]. The sera of control and KO *Casr* mice were collected and assayed for their intact PTH (1–84) content, revealing levels ~17-fold higher than those in control mice, due to the lack of functional CaSR (Fig. 4b). Two parathyroid glands from each mouse were dissected free of the surrounding thyroid and connective tissues and then incubated for 1 h in different buffers containing increasing concentrations of Pi in high and low $Ca_o^{2+}$ (Fig. 4c). Parathyroid glands from the control mice exhibited significantly increased PTH secretion in response to pathophysiological Pi concentrations. In contrast, in glands from the KO *Casr* mice in which CaSR had been functionally knocked out, Pi-mediated stimulation of PTH secretion was not observed (Fig. 4c).

**CaSR anion-binding sites are conserved across species.** Four different multivalent anion-binding sites (1–4), hereafter termed Pi-binding sites, were identified in the inactive and active CaSR-ECD structures (Fig. 5)[27]. We examined the level of amino acid conservation at the predicted Pi-binding sites, by using the phylogenetic family C GPCR alignment of Herberger et al.[29]. Herberger's data set comprised 138 nucleotide sequences including 42 CaSR sequences, from different phylogenetic groups. WebLogo revealed higher bit scores at the predicted residues involved in Pi binding in the vertebrate CaSR subgroup, corresponding to strong amino acid conservation, compared with all collected family C GPCR sequences (Fig. 6). The four CaSR-predicted Pi-binding sites are characterized by an abundance of positively charged residues (here represented with blue), with arginine being particularly conserved. Interestingly, predicted binding sites 1–3 are more conserved within the CaSR subgroup as illustrated in the WebLogo analysis (Fig. 6). The CaSR

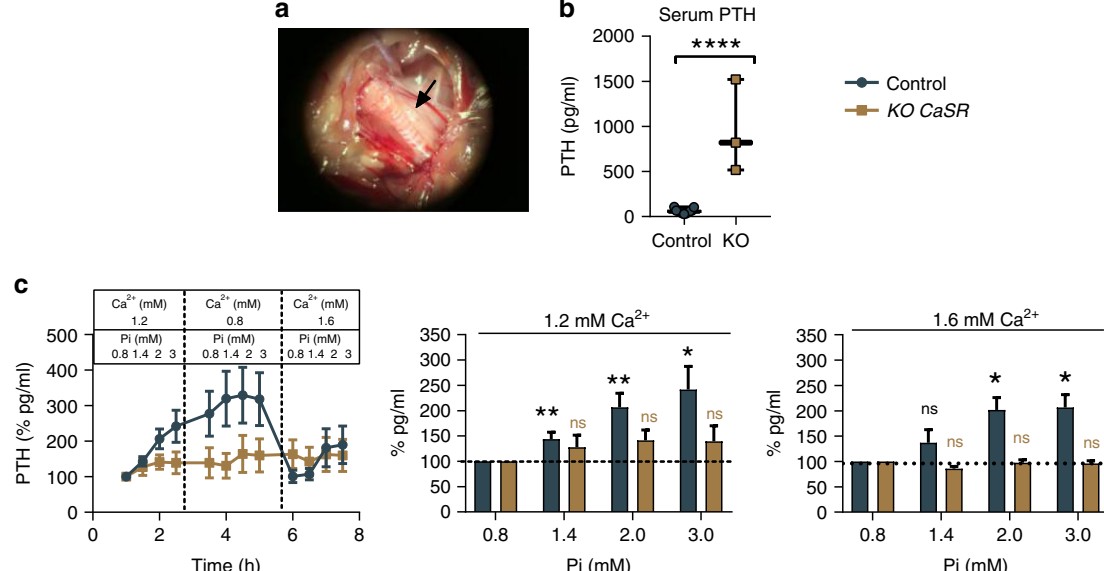

**Fig. 4** Pi increases murine PTH secretion via the CaSR. **a** Magnified mouse parathyroid gland before dissection in a 10-day-old control mouse. **b** Serum PTH levels measured from control and KO *Casr* mice before dissection. Data shown as median and min. to max. range (WT $N = 11$ and KO *Casr* $N = 3$). [****]$P < 0.0001$ by unpaired $t$ test. **c** Effect of pathophysiologic Pi exposure at different $Ca^{2+}$ concentrations on PTH secretion in freshly isolated parathyroid glands from 7- to 10-day-old control. WT glands (from four different litters) were sequentially exposed to normal $Ca^{2+}$ (1.2 mM $Ca^{2+}$; $N = 11$), low $Ca^{2+}$ (0.8 mM $Ca^{2+}$; $N = 7$ and 0.6 mM; $N = 4$), and high $Ca^{2+}$ (1.6 mM $Ca^{2+}$; $N = 7$). The KO *Casr* glands were exposed to 1.2, 0.8, and then 1.6 mM $Ca^{2+}$ in turn ($N = 3$). PTH secretion measured every 30 min and normalized to baseline conditions (0.8 mM Pi/1.2 mM $Ca^{2+}$) in the time course (left). For bar graphs, data normalized to 0.8 mM Pi at their respective $Ca^{2+}$ concentration. Data shown as %mean ± SEM. ns not significant, [*]$P < 0.05$, [**]$P < 0.01$ by RM-ANOVA (Dunnett's multiple comparison test). Source data are provided as a Source Data file

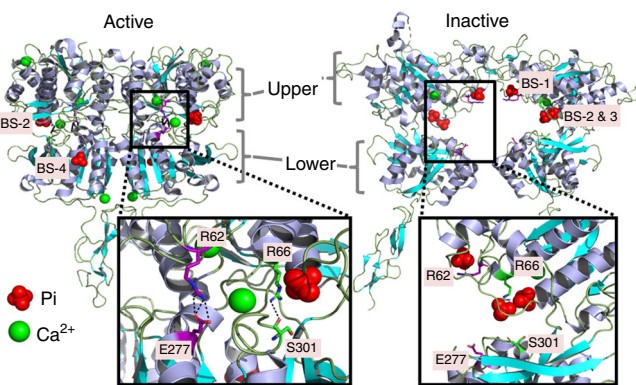

**Fig. 5** Pi-binding sites in the CaSR extracellular domain. Pi-binding sites in the CaSR's active and inactive conformations. In the active conformation (left), R62 and R66 interact respectively with E277 (creating a salt bridge) and S302 (hydrogen bond) to keep the upper and lower domain closely associated ensuring a closed/active conformation. In the inactive conformation (right), both interactions are broken and Pi ions stabilize the positive charges of R62 and R66. Also, Pi ions displace a lower domain eliciting an open/inactive conformation, by reducing equilibrium free energy. Note that Pi-binding site 1 is only occupied in the inactive state, whereas Pi-binding site 2 is occupied in both inactive and active conformations integrated within the receptor's structure. Pink sticks show the R62–E277 salt bridge interaction and green sticks show the R66–S301 hydrogen bond. PDB accession numbers 5k5s and 5k5t[27]. BS binding site

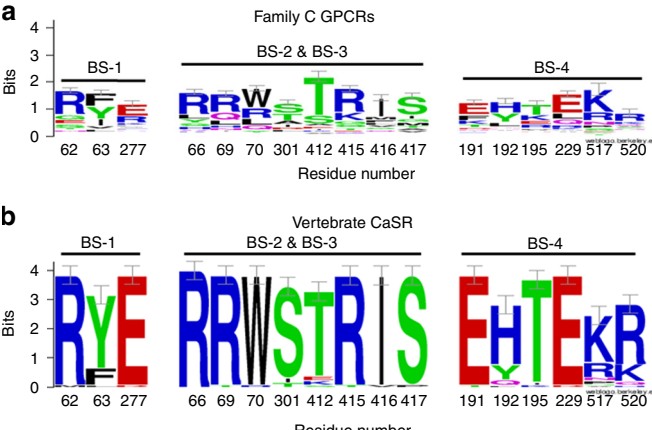

**Fig. 6** CaSR Pi-binding sites are conserved across species. Phylogenetic analysis of the four Pi-binding sites within the CaSR ECD, represented with WebLogo and following the human numbering scheme[27]. The data set comprised 138 aligned nucleotide sequences from the family C GPCRs, including 42 CaSR sequences from different phylogenetic groups[29]. **a** Aligned amino acid positions from family C GPCRs, including CaSR. **b** Vertebrate CaSR subset reveals overall higher bit scores, reflecting stabilization and conservation of Pi-binding sites. Conservation at each amino acid position in the binding site is assessed as the WebLogo bit score, with amino acid usage frequencies at that position reflected in the relative size of the single-letter amino acid identifiers within the bit score bars. BS binding sites

subgroup analysis revealed high conservation of Pi-binding sites across the 42 different species with diverse evolutive origin. This analysis further confirmed that the residues involved in the Pi-binding sites which are present in the inactive conformation, including R62 and E277 (site 1), and R66 and S301 (site 3) are highly conserved (Fig. 6).

To identify the site of Pi-mediated CaSR inhibition, we focused on the two inactive conformation-specific Pi-binding sites 1 and 3 (Fig. 5). In the absence of Pi, R62 and E277 (binding site 1) form a salt bridge across the upper and lower domains of the Venus flytrap that stabilizes the CaSR's active conformation by promoting ECD closure (Supplementary Movie 1). When Pi is present, this salt bridge is broken and the positive charge of R62 is stabilized by direct interaction with the negative charges of Pi. By breaking the salt bridge, Pi displaces E277 on the lower domain, thus reducing the free energy needed to shift the equilibrium toward the inactive (open) conformation of the receptor (Fig. 5). In binding site 3, when Pi is absent, R66 interacts with S301 forming a hydrogen bond across the upper and lower domains that also promotes receptor closure and this stabilizes the CaSR's active conformation. In contrast, Pi breaks the hydrogen bond by interacting directly with R66 and partially with R69, which also contributes to the inactive conformation by displacing S301 in the lower domain (Fig. 5). In addition, $Ca^{2+}$ binding site 3 stabilizes the loop conformation that supports formation of the inter-domain bond between R66 and S301. These features highlight the importance of $Ca^{2+}$ binding site 3 and Pi-binding sites 1 and 3 in regulating CaSR's activity.

**Pi-mediated CaSR inhibition requires residue R62.** Based on the in silico analysis, we selected residues R62 and R66 as candidates for mutagenesis to determine their possible contribution to Pi-mediated CaSR inhibition. These residues were mutated separately to alanines ($CaSR^{R62A}$ and $CaSR^{R66A}$) and the CaSR constructs arising were expressed transiently in wild-type HEK-293 cells. Immunoblotting revealed successful expression of the mutated receptors at levels similar to the WT receptor (Supplementary Fig. 7a). We expected decreased sensitivity of $CaSR^{R62A}$ and $CaSR^{R66A}$ to $Ca_o^{2+}$ due to likely disturbances of Venus flytrap closure. This was confirmed experimentally whereby the potencies of $Ca_o^{2+}$-induced $Ca_i^{2+}$ mobilization for these mutants were reduced ($EC_{50}$ for $Ca_o^{2+}$, 3.4 mM WT; 5.2 mM $CaSR^{R62A}$; 5.3 mM $CaSR^{R66A}$). The $Ca_o^{2+}$ $E_{max}$ values for these mutants, however, were not altered (Supplementary Fig. 7b).

We next evaluated the effect of pathophysiologic Pi concentrations on CaSR-induced $Ca_i^{2+}$ mobilization in $CaSR^{R62A}$ and $CaSR^{R66A}$. It was found that 2 mM Pi had no effect on $CaSR^{R62A}$-induced $Ca_i^{2+}$ mobilization upon stimulation with different concentrations of $Ca^{2+}$ (Fig. 7a) or cotreatment with R568 (Fig. 7b). Due to the mutants' decreased sensitivity to $Ca^{2+}$, higher $Ca^{2+}$ concentrations were used (1 mM $Ca^{2+}$ for $CaSR^{R62A}$ and 1.5 mM $Ca^{2+}$ for $CaSR^{R66A}$) to achieve equivalent responses to $CaSR^{WT}$ (0.5 mM $Ca^{2+}$) (Fig. 7b and Supplementary Fig. 9). Similarly, $CaSR^{R62A}$ was also insensitive to 2 mM $SO_4$ (Fig. 7c). $CaSR^{R62A}$ responses were also identical to time-matched controls when exposed to either 1 or 1.5 mM $Ca_o^{2+}$, in the presence of 1 μM R568, with or without Pi (Supplementary Fig. 8a,b). Cell surface receptor expression levels were also evaluated and confirmed to be equivalent for $CaSR^{WT}$ and $CaSR^{R62A}$, and significantly greater than those for the well-known trafficking mutants, E799A and I642Del (Fig. 7d). Next, we measured $Ca_i^{2+}$ mobilization in response to increasing $Ca^{2+}$ concentration (0.5–5 mM), in the presence of R568, in $CaSR^{R62A}$-stably transfected HEK-293 cells ($CaSR^{R62A}$-HEK) and found that $E_{max}$ values were not altered in the presence of Pi ($P > 0.05$) (Fig. 7e). We did note that the $EC_{50}$ values for $Ca_i^{2+}$ mobilization in response to $Ca_o^{2+}$ were significantly reduced in the presence of Pi, from 2.6 to 1.6 mM (in 0.8 mM Pi) and 1.7 mM (2 mM Pi), showing partial recovery of the loss of function. There were no

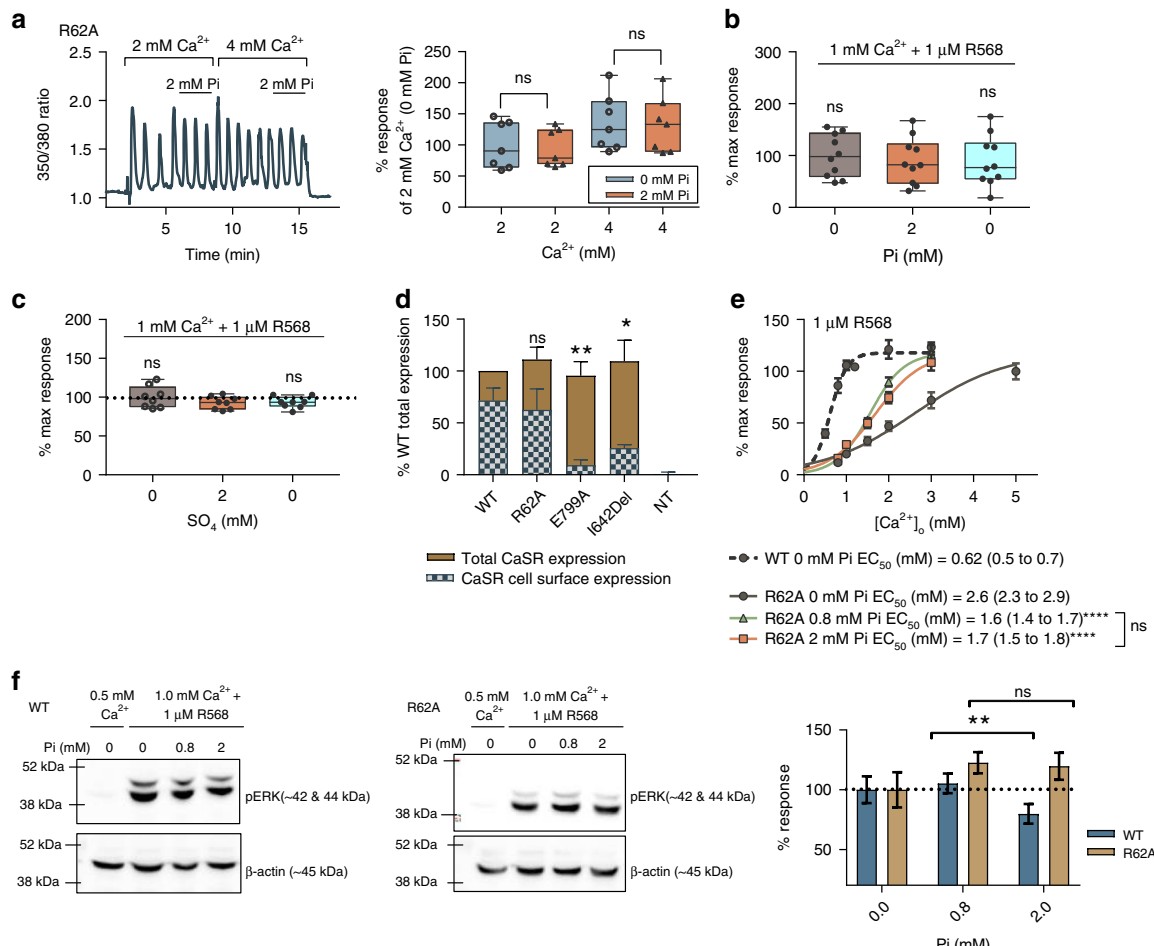

**Fig. 7** $CaSR^{R62A}$ is not inhibited by Pi. $CaSR^{R62A}$-induced $Ca_i^{2+}$-mobilization is not sensitive to Pi upon stimulation with $Ca^{2+}$ (**a**, $n = 7$), or cotreatment with R568 (**b**, $n = 10$). **c** $CaSR^{R62A}$-induced $Ca_i^{2+}$-mobilization is also insensitive to $SO_4$ ($n = 8$). **a–c** Area under the curve was calculated for each treatment and normalized against maximal response. **d** Cell surface expression levels of $CaSR^{WT}$, $CaSR^{R62A}$, and known defective mutants measured in transiently transfected HEK cells with non-transfected HEK-293 cells (NT) used as negative controls ($n = 9$, performed in triplicate; NT, $n = 6$). Data normalized to $CaSR^{WT}$ (WT) total expression levels. **e** $Ca^{2+}$ concentration−effect curves for $Ca_i^{2+}$-mobilization on $CaSR^{R62A}$ (with 0, 0.8, or 2 mM Pi) and $CaSR^{WT}$ (without Pi, shown here for comparison purposes). Data fitted to a four-parameter Hill equation (Eq. (1)) for sigmoidal concentration−response variable slope ($n = 8$–10, from three independent experiments). Data fitted best when $EC_{50}$ (shown as mean with 95% confidence interval) values for $Ca_o^{2+}$ were different among data sets and $E_{max}$ was shared, $P < 0.01$ extra sum-of-square $F$-test. **f** Representative immunoblot showing pERK responses by $CaSR^{R62A}$ and $CaSR^{WT}$ after a 10-min stimulation; 0.5 mM $Ca^{2+}$ used as negative control. Changes in pERK were corrected for β-actin abundance and then normalized to the Pi-free stimulated control ($n = 8$, from four independent experiments). Data are plotted as %mean ± SEM and analyzed by using RM-ANOVA, Dunnett's multiple comparisons (**b–e**), or unpaired $t$ test (**a**, **f**). ns not significant, *$P < 0.05$ and **$P < 0.01$. Source data are provided as a Source Data file

significant differences between 0.8 and 2 mM Pi responses in either potency ($EC_{50}$) or efficacy ($E_{max}$) (Fig. 7e).

The effect of Pi on $CaSR^{R62A}$ was also studied over a more extended range of Pi concentrations (0−5 mM) (Supplementary Fig. 8c). Again in $CaSR^{R62A}$, physiologically relevant Pi concentrations (0.8−1.4 mM) increased $Ca_o^{2+}$-induced $Ca_i^{2+}$ mobilization, when compared with 0 mM Pi (control). Nevertheless, the inhibitory effect of Pi was absent in $CaSR^{R62A}$ at any of the Pi concentrations tested (up to 5 mM Pi) (Supplementary Fig. 8c). Next, by using a separate readout of receptor activity increased Pi concentration failed to inhibit pERK in $CaSR^{R62A}$-HEK cells, whereas in wild-type CaSR-HEK cells the pERK response was significantly reduced (by up to 30%) as before (Fig. 7f).

In contrast, we found that $CaSR^{R66A}$ exhibited Pi-induced inhibition of $Ca_o^{2+}$-induced $Ca_i^{2+}$-mobilization that closely resembled responses by $CaSR^{WT}$ (Supplementary Fig. 9a). In

the presence of 2 mM Pi, the maximal response of $CaSR^{R66A}$ to $Ca_o^{2+}$ ($E_{max}$) was reduced significantly (by up to 60%), whereas the $EC_{50}$ for $Ca_o^{2+}$ was unchanged (Supplementary Fig. 9b). Together, these data implicate R62 (Pi-binding site 1) as the key residue responsible for Pi-induced CaSR inhibition.

## Discussion

Here we identify a mechanism for sensing serum Pi via the CaSR. By combining in vitro signaling experiments with ex vivo PTH secretion measurements from human and KO *Casr* murine parathyroid glands, we reveal that Pi acts as a noncompetitive antagonist of the CaSR at pathophysiological concentrations, resulting in increased PTH secretion. This molecular mechanism may explain the stimulatory effect of Pi on PTH secretion under physiological conditions and in SHPT.

We first characterized the effects of Pi on CaSR-mediated signaling, demonstrating that raising Pi concentration from 0.8

(physiological) to 2 mM (CKD-like) significantly inhibited maximal levels of $Ca_i^{2+}$-mobilization by 45% and pERK by 30%. Experiments were designed to limit Ca×P product formation primarily by employing low-to-normal $Ca_o^{2+}$ concentrations (from 0.5 to 1.2 mM) in the presence of CaSR PAMs, namely the calcimimetics R568, cinacalcet and spermine, to potentiate CaSR activation but using less $Ca^{2+}$. Adding 2 mM Pi to the experimental buffer decreased ionized $Ca^{2+}$ concentration by <7%. However, even when the $Ca_o^{2+}$ concentration was increased to correct this small Pi-mediated decrease in free $Ca^{2+}$, Pi still inhibited CaSR activity. In addition, lowering buffer pH to 7.2, equivalent to metabolic acidosis seen in some CKD patients[17], failed to overcome the inhibitory effect of Pi, even though it would be expected to reduce the risk of Ca × P formation. These data were consistent with previous observations that pathophysiologic acidosis itself inhibits CaSR[30,31]. Together, these findings exclude the possibility that the CaSR-inhibiting effect of Pi seen here can be explained by its potential to lower ionized $Ca^{2+}$ concentration. Regarding CaSR-induced $Ca_i^{2+}$-mobilization, Pi significantly inhibited receptor efficacy, reducing the $E_{max}$ by 37 ± 3% at 2 mM Pi, without affecting potency ($EC_{50}$) for $Ca_o^{2+}$. Thus, Pi-mediated CaSR inhibition could not be overcome by using higher $Ca_o^{2+}$ concentrations, suggesting a noncompetitive mechanism. Further, Pi attenuated CaSR activity with an $IC_{50}$ of ~1.2 mM, confirming that the CaSR is sensitive to Pi within the physiological range (0.8–1.4 mM). The actions of Pi at concentrations >1.5 mM are relevant for CKD whose cardinal features include hyperphosphatemia (>2 mM). Indeed, when CaSR was stimulated with cinacalcet, a drug widely used in dialysis patients, 2 mM Pi reduced the CaSR response by 46 ± 8%. Therefore, variations in serum Pi concentration might explain some of the variability of cinacalcet effectiveness observed across CKD patients[19,25], though it should be noted that these values were obtained by using CaSR-HEK cells, whereas CaSR sensitivity for $Ca^{2+}$, R568, and Pi differs in parathyroid cells.

Consistent with CaSR-ECD structural data[27], $SO_4$ also attenuated CaSR activity in the same range as Pi; however, as physiological $SO_4$ concentrations are lower (~0.3 mM in serum), $SO_4$ is less likely to be a physiologically relevant CaSR inhibitor. On the other hand, plasma $SO_4$ levels can also be elevated in CKD and therefore might also contribute to SHPT[32].

The association of increased Pi concentration with both excessive PTH secretion and parathyroid gland hyperplasia has been widely demonstrated in vitro and in vivo although the causal link remains uncertain[4,7,8,15,21,33,34]. We tested our hypothesis by measuring PTH secretion from freshly isolated human parathyroid cells ex vivo in a perifusion model. PTH secretion experiments were usually conducted within 24 h (maximum 48 h) from surgical excision and tissue was stored undigested at 4 °C to minimize the loss of CaSR expression reported in previous studies[35]. Parathyroid cell preparations were initially exposed to low $Ca^{2+}$ concentrations to induce PTH secretion and to confirm CaSR expression and its functional responsiveness. Then, the parathyroid cells were exposed to small increases in Pi concentration, within the CKD range, and a rapid (within 2 min) and quickly reversible rise in PTH secretion consistent with a cell surface receptor-mediated action was observed that cannot be explained by genomic changes alone. These results are consistent with a previous in vivo study performed in rats in which duodenal or intravenous Pi injections elicited rapid (within 10 min) increases in plasma PTH, and the authors hypothesized the existence of a cell surface "phosphate sensor", similar to the $Ca^{2+}$ sensor, and expressed in the duodenum and the parathyroid gland[36]. An alternative hypothesis for the progression of SHPT is that partial downregulation of parathyroid CaSR expression occurs,

permitting increased PTH secretion and this would mean fewer CaSRs for Pi to inhibit. However, one of the leading treatments for SHPT in advanced renal disease are the calcimimetics (cinacalcet and etelcalcetide), which target the CaSR directly and provide clinically significant reductions in PTH secretion. Therefore, even in advanced CKD, there remains sufficient CaSR expression to regulate PTH secretion[15,37]. Furthermore, Pi also rapidly stimulated PTH secretion in control mice parathyroid glands, even though physiological serum Pi levels in mice are higher than those in humans (+0.6 mM)[38]. In contrast, in parathyroid-specific KO Casr mice, which lacked $Ca_o^{2+}$-sensitivity, the Pi inhibitory effect was markedly impaired at each of the $Ca_o^{2+}$ concentrations tested, indicating a direct effect of Pi on the parathyroid glands mediated by the CaSR. It should be noted that parathyroid glands from KO Casr mice exhibit elevated PTH secretion and therefore the potential for further substantial increases in secretion may be impaired.

The idea that CaSR is a Pi sensor is supported by the crystal structures of the human CaSR ECD, obtained by Geng et al., which revealed four multivalent anion-binding sites[27]. However, a different CaSR-ECD crystal structure obtained by Zhang et al. did not reveal any potential Pi-binding sites, although Pi was not present in the crystallization buffer[39]. From Geng's crystal structure model, we assessed which putative anion-binding site(s) were most likely to mediate functional inhibition, based on their presence in the inactive and active conformations. Anion-binding site 1, based in part on R62, and site 3, based in part on R66, were present solely in the inactive conformation. Thus, we focused on R62 and R66, which because of their positively charged side chains interact directly with Pi ions. The loss of function displayed by $CaSR^{R62A}$ and $CaSR^{R66A}$ can be explained by impairment of R62–E277 and R66–S301 interactions occurring between the upper and lower domain, as they both support receptor stability in the closed (active) conformation. The crucial finding, however, was that R62 appears to be the principal residue accounting for anion-dependent inhibition. Specifically, $CaSR^{R62A}$ exhibited no anion-induced inhibition of CaSR in either $Ca_o^{2+}$-induced $Ca_i^{2+}$-mobilization or pERK assays performed on $CaSR^{R62A}$-HEK cells. The R62–E277 salt bridge that connects upper and lower domains, holds the receptor in its active conformation (Supplementary Movie 1). This interaction is disrupted upon Pi binding to the positive charge of R62, releasing free energy and displacing the receptor equilibrium toward the inactive conformation. Despite the salt bridge being a noncovalent interaction, its ability to increase stability and reduce entropy of the folded conformation of proteins is well known[40]. Therefore, mutation of R62 will break this salt bridge, thus mimicking the effect of Pi and so preventing any further inhibitory action, consistent with the loss of function and lack of Pi-mediated inhibition in $CaSR^{R62A}$ seen here. In addition, that Pi fails to inhibit $CaSR^{R62A}$ further demonstrates that Pi acts directly on the CaSR rather than via Ca × P formation. Indeed, $CaSR^{R62A}$ activity increased with low concentrations of Pi, possibly by supporting the loop conformation that allows $Ca^{2+}$ binding in the interdomain cleft (site 3)[27], which is impaired in this mutant and enhances domain closure for receptor activation. However, increased CaSR activity with low Pi was not seen in $CaSR^{WT}$, as $Ca^{2+}$ binding site 3 and R62–E277 already stabilize this loop conformation. In contrast, $CaSR^{R66A}$ was inhibited by Pi in the same way as $CaSR^{WT}$, pointing to a primary inhibitory role of R62 upon Pi binding, and a primary role of R62–E277, rather than R66–S301, in stabilizing the active conformation of the receptor. Further, the CaSR Pi-binding sites exhibit a high degree of amino acid sequence conservation across species, largely absent from other family C GPCRs, suggesting an important role for these sites in CaSR biology.

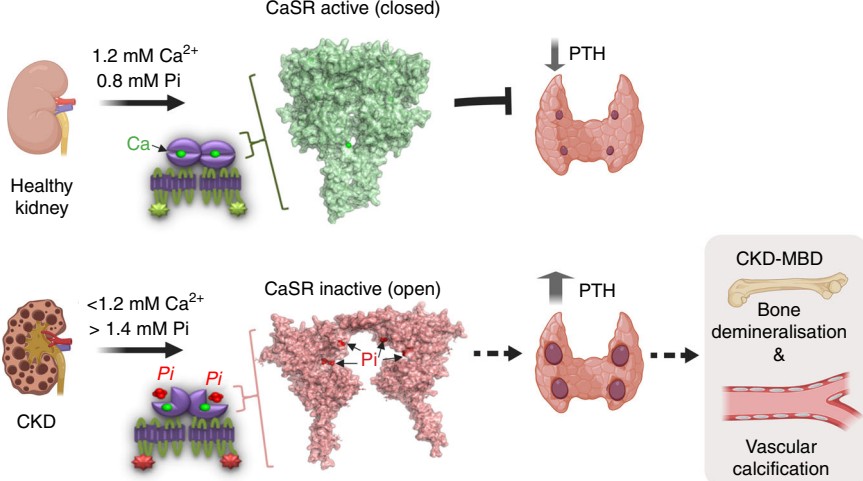

**Fig. 8** Pi is an important regulator of CaSR activity. In health, CaSR negatively modulates PTH secretion to control $Ca^{2+}$ homeostasis. An acute rise in Pi inhibits the CaSR and would permit increased PTH secretion which stimulates renal Pi excretion to help restore normal serum Pi levels. However, in the hyperphosphatemia of CKD, the additional Pi would stabilize the CaSR's inactive conformation permitting chronically elevated PTH secretion and contributing to SHPT progression. CaSR-ECD crystal models obtained from PDB 5k5s (active) and 5k5t (inactive) and shown as surface. Created with Biorender.com

We have demonstrated that the CaSR senses small deviations in Pi concentration over the human pathophysiological range. Pi, acting at residue R62, is a noncompetitive antagonist of the CaSR, and in this respect can be considered an endogenous calcilytic permitting increased PTH secretion with relevance for the physiology of $Ca^{2+}$ homeostasis and for CKD. We propose that Pi regulates the active–inactive equilibrium of CaSR such that at higher plasma Pi concentrations, the CaSR equilibrium is displaced toward its inactive conformation, thus permitting increased PTH secretion (Fig. 8). In turn, elevated plasma PTH levels stimulate $Ca^{2+}$ reabsorption from the kidney, release $Ca^{2+}$ and Pi from the bone, and then restore normal serum Pi concentration via enhanced proximal tubule Pi wasting in the kidneys. Therefore, Pi might represent a tonic modulator of $Ca^{2+}$-induced CaSR activation permitting fine-tuning of the calciotropic response depending on the ambient serum Pi concentration. Moreover, because the CaSR additionally regulates PTH synthesis and parathyroid cell proliferation, this mechanism may also contribute to the effect of Pi on PTH mRNA stability and parathyroid hyperplasia[41].

In the context of CKD, where serum Pi levels are increased due to the impaired renal excretion, the homeostatic response described above could produce continuously elevated PTH secretion leading to SHPT, bone loss, and further hyperphosphatemia. In addition to hyperphosphatemia, there are other factors that also contribute to the development and progression of SHPT in the context of CKD: low $1,25OH_2D$, hypocalcemia, elevated levels of FGF23 and Klotho, and reduced levels of vitamin D receptor[11,15,42–44]. FGF23 production in osteocytes and osteoblasts is stimulated by Pi and $1,25OH_2D$ and acts on the proximal tubule to promote Pi wasting[15]. There is a clear association between SHPT progression and serum FGF23 levels given the interplay between the PTH/PTH1R and FGF23/Klotho pathways on Pi homeostasis; however, the specific contributions of each regulatory pathway have not been fully elucidated[15,45]. Several agencies, including the US National Kidney Foundation and the UK Renal Association, have established recommended values for serum PTH, $Ca^{2+}$, $1,25OH_2D$, and Pi levels in the management of SHPT for CKD patients[22–24]. According to the UK Renal Association, only 30% of CKD patients have serum $Ca^{2+}$, PTH, and Pi levels simultaneously under control, and only 60% have serum Pi levels within the recommended range (1.1–1.7 mM)[22]. Therefore, it will be interesting to determine whether strategies aimed at reducing extracellular Pi concentrations in CKD patients with SHPT might enhance the effectiveness of calcimimetics (cinacalcet, evocalcet, and etecalcetide) in lowering serum PTH levels.

There is also a possibility that Pi may elicit a protective role in patients with autosomal-dominant hypocalcemia. These patients express activating mutations in CaSR, or in G protein 11 (GNA11), which cause hypocalcemia, inappropriately low PTH levels, and moderate hyperphosphatemia[46,47]. Therefore, such hyperphosphatemia might moderate the gain of function by partially inhibiting the CaSR, so limiting the severity of the hypocalcemia, hypercalciuria, and nephrocalcinosis. Indeed with CaSR expressed in numerous tissues throughout the body, our findings may provide a more general Pi-sensing mechanism for the maintenance of extracellular Pi, for example in CaSR-expressing osteoblasts or gut cells where Pi has direct effects.

In summary, these results provide an original molecular mechanism that explains Pi-stimulated PTH secretion and further provide a mechanism for sensing elevated Pi concentrations in CaSR-expressing tissues. Then in the hyperphosphatemia of CKD, Pi-mediated stabilization of the CaSR's inactive conformation would elicit chronically increased PTH release as the damaged kidneys fail to excrete the Pi, leading to SHPT, and in turn to further bone loss and hyperphosphatemia (Fig. 8).

## Methods

**Cell culture**. HEK-293 cells (from ATCC) were used as a model for the functional expression of CaSR protein. Wild-type HEK-293 and stably transfected CaSR-HEK-293 (CaSR-HEK) cells were grown in Dulbeccos' modified Eagle's medium containing 1.8 mM $CaCl_2$ and 0.9 mM $Na_2HPO_4$ from Sigma (D5796), and fortified with 10% fetal bovine serum (FBS). Cells were dissociated with trypsin upon reaching 80–90% confluence and grown at 37 °C with 5% $CO_2$ until passage 30.

**ERK activation assay**. CaSR-HEK cells were grown on 30-mm dishes and then incubated at 37 °C in experimental buffer containing (mM) 20 HEPES (pH 7.4), 125 NaCl, 4 KCl, 0.5 $CaCl_2$, 0.5 $MgCl_2$, and 5.5 glucose with Pi added as $Na_2HPO_4$ and $KH_2PO_4$ in a 4:1 ratio (pH 7.4). Following equilibration (20 min), cells were treated for 10 min with experimental buffer including different Pi concentrations and then lysed on ice for 15 min in a RIPA-like buffer supplemented with protease

and phosphatase inhibitors ((mM) 12 HEPES (pH 7.4), 300 mannitol, 1 EGTA, 1 EDTA, 1% (v/v) Triton-X 100, 0.1% (w/v) sodium dodecyl sulfate (SDS), 1 NaF, 250 μM sodium pyrophosphate, 100 μM sodium vanadate, 1.25 μM pepstatin, 4 μM leupeptin, 4.8 μM PMSF, and 1 N-ethylmaleimide). Phosphorylation of ERK (pERK) was quantified by semiquantitiative immunoblotting.

**Immunoblotting protein analysis**. Protein samples were first diluted (4:1) with 5× Laemmli buffer followed by resolution of the proteins by SDS-PAGE and electrophoretic transfer into a nitrocellulose membrane (GE Healthcare Life Sciences). Membranes were incubated in blocking buffer solution containing Tween-TBS solution and 5% (w/v) bovine serum albumin (BSA) for 1 h and probed with primary antibodies for pERK [Phospho-p44/42 MAPK (Erk1/2) (Thr202/Tyr204) (E10) Mouse (1:3000 dilution) from Cell Signaling (Cat. number 9106)], CaSR [Calcium Sensing Receptor monoclonal mouse antibody (5C10, ADD) (1:3000) from Thermo Fisher Scientific (Cat. number MA1-934)], or β-actin [β Actin-Peroxidase monoclonal (AC-15) mouse antibody (1:25,000) from MERCK (Cat. number A3854)] for 1 h. Secondary antibody was horseradish peroxidase-conjugated horse anti-mouse IgG (1:10,000) from Cell Signaling (Cat. number 7076). Chemiluminescence was generated by using BIO-RAD clarity western ECL substrate, detected by using a Chemidoc gel analyzer (BIO-RAD), and quantified by densitometry by using Image Lab 6. Uncropped and unprocessed immunoblot scans shown in Figs. 2a and 7f and in Supplementary Fig. 7a are supplied in the Source Data file.

**Site-directed mutagenesis**. Alanine mutations were introduced into the human wild-type CaSR gene by using QuickChange Lightning Site-directed mutagenesis kit (Stratagene). Mutations were confirmed by the Genomic Technologies Core Facility at The University of Manchester. HEK-293 cells were then transiently transfected with wild-type or mutant receptors by using FuGENE-6 (Roche Diagnostics). Cells were used experimentally within 72 h of transfection. For stable transfection, vectors were first linearized (Ssp1) and then CaSR-expressing cells were selected by using hygromycin (400 μg/ml).

**Ionized calcium measurement**. Free ionized $Ca^{2+}$ was measured by using a GEM premier 5000 gas analyzer with an ion-selective electrode by the Clinical Biochemistry department of the Manchester Royal Infirmary.

**Intracellular calcium mobilization imaging**. Dual-excitation-wavelength microfluorometry was carried out on cells cultured on 13-mm glass coverslips for 48 h in DMEM with 10% FBS. Cells were loaded with 1 μM FURA2-AM in 1.2 mM $Ca^{2+}$ experimental buffer supplemented with 0.1% (w/v) BSA for 1–2 h at room temperature in the dark. Experimental buffer contained (mM) 0.5 $Ca^{2+}$, 20 HEPES (pH 7.4), 118–125 NaCl, 0.5 $MgCl_2$, 4 KCl, and 5.5 glucose supplemented with Pi as $Na_2HPO_4$ and $K_2HPO_4$ added in a 4:1 ratio. Cells were mounted in a perfusion chamber and observed through ×40 oil-immersion lens on a Nikon Diaphot inverted microscope.

**Receptor cell surface assay**. HEK-293 cells were transiently transfected with pcDNA3.1(+) vector (Life Technologies, USA) containing CaSRs that included a Flag tag inserted after the signal peptide. Cells were then grown in a 96-well plate until 80–90% confluent, then washed with Tween-TBS buffer (15 mM Tris (pH 8.0), 150 mM NaCl, and 0.1% (v/v) Tween 20), and then fixed with either 100% methanol (total expression of CaSR) or 4% paraformaldehyde (receptor cell surface expression levels) for 15 min at 4 °C. Following blocking with 1% skimmed milk (1 h), cells were incubated for 1 h with mouse Anti-Flag antibody H2-peroxidase (A8592, Sigma-Aldrich, USA). Excess antibody was removed by washing with Tween-TBS and 3,3′,5,5′-tetramethylbenzidine (TMB) substrate added with the peroxidase reaction stopped after 10 min by using 1 M HCl. Absorbance was measured at 450 nm.

**Origin and preparation of human parathyroid cells**. Normal human parathyroid tissue was obtained with consent from patients undergoing total thyroidectomy for major thyroid disease (e.g., large multinodular goiters or Graves' disease) when the surgeon identified and removed one parathyroid gland, subjected it to mincing, and then returned it via a syringe to the ipsilateral sternomastoid muscle, to reduce postoperative hypoparathyroidism[48]. Any parathyroid gland tissue fragments left in the syringe were provided under informed consent and de-identified from the Royal North Shore Hospital (St. Leonards, NSW, Australia) and the Mater Misericordiae Hospital (North Sydney, NSW). Approval for access to human parathyroid tissue was provided by the Human Research Ethics Committee of St. Vincent's Health Network, Darlinghurst, NSW 2010 (SVH File Number 11/067). We confirm that we have complied with all relevant ethical regulations for research.

The parathyroid specimen was transported in ice-cold Hanks' balanced salt solution (Invitrogen) containing 1.25 mM $CaCl_2$ and stored as sliced tissue cubes at 4 °C in MEM (11380-0.37, Invitrogen) for up to 48 h. Digestion of the tissue was performed as follows. First, the specimen was incubated at 37 °C for 20 min in a MEM-oxygenated solution containing 1 mg/ml collagenase (Type I, Worthington,

Scimar, Victoria) and 0.2 mg/ml DNase I (Type IV, Sigma). After 20 min, the enzyme solution was decanted and the parathyroid tissue transferred into fresh MEM containing 1 mg/ml bovine serum albumin (BSA, protease-free, Sigma A-3059). The tissue was then triturated by repeated passage through the tip of a disposable 5-ml syringe (without a needle). The cloudy suspension containing clumps of parathyroid cells was filtered with a 200-μm pore-size nylon filter and centrifuged for 2 min at $100 \times g$ (Sorvall H-1000B rotor). The pellet was resuspended and washed twice in 5 ml of MEM supplemented with 1 mg/ml BSA. The remaining undigested tissue was returned to MEM medium containing collagenase and DNase I and incubated for a further 20 min at 37 °C followed by additional trituration and centrifugation as before. The first and second cell harvests were combined for analysis[49,50].

**Human PTH assay**. Parathyroid cells obtained from human parathyroid tissue were used immediately after tissue disaggregation and digestion. Parathyroid cells (~20,000–50,000) were loaded onto the cell surface of a 1-ml bed volume of Bio-Gel P-4 (nominal exclusion limit 4 kDa) and then gently covered with a 1-ml bed volume of sephadex G-25 (nominal exclusion limit 5 kDa) in a small perifusion column. The column was equilibrated for 20 min by using (mM) 125 NaCl, 4 KCl, 1.25 $CaCl_2$, 1 $MgCl_2$, and 20 HEPES (pH 7.4) buffer supplemented with 0.1% D-glucose 1× basal amino acid mixture, 1 mg/ml BSA, as well as with 0.8–3 mM $Na_2HPO_4$ and $K_2HPO_4$ added in a 4:1 ratio. Cells were perifused (at 37 °C and 1.5 ml/min) by gel filtration with intact PTH (~9 kDa) detected in the void volume. Tubing connections were established downstream to a peristaltic pump and upstream to a reservoir. Samples were collected every 2 min (3 ml) into chilled tubes and then frozen (−80 °C) until analysis for intact human PTH by using a third-generation, two-site chemiluminescence assay on an Immulite 2000 auto-analyzer (Diagnostic Products Corporation, CA). As required, perifusion solutions were changed to allow variations in $Ca^{2+}$ and Pi concentrations in experimental buffers.

**Murine parathyroid gland isolation and ex vivo culture**. Mice with homozygous parathyroid gland-specific deletion of exon 7 (encoding the seven-transmembrane domain, three intracellular loops, and intracellular tail) of the CaSR (hereafter termed KO *Casr*) were generated cross-breeding of floxed *Casr* mice with mice expressing parathyroid-specific Cre[28,51]. Littermates that did not express both transgenes simultaneously were used as controls. Two parathyroid glands from each mouse were dissected free of the surrounding thyroid and connective tissues and submerged in a microdroplet (10 μl) of secretion medium [MEM Eagles with Earle's balanced salts supplemented with 0.5 mM $Mg^{2+}$, 0.2% bovine serum albumin, and 20 mM HEPES (pH 7.4)] and placed in the middle of a 13-mm track-etched (0.1 μM pore) polycarbonate membrane (Whatman), floating on a large drop (500 μl) of ice-cold secretion media supplemented with 3 mM $Ca^{2+}$ [51]. The time used to find, dissect, and clean the parathyroid glands from each mouse was <5 min per animal. The dissected glands were equilibrated in fresh secretion media (at 37 °C) for 1 h. Then, each pair of glands were incubated for 30 min in 500 μl of secretion media (37 °C) containing different concentrations of $Ca^{2+}$ and Pi. Intact PTH (1–84) secreted into the bathing culture media was then collected and quantified by ELISA assay (Immunotopics Inc.) and expressed as pg PTH/30 min/gland. Mouse serum was also collected and assayed for intact PTH. Only KO *Casr* mice displaying high serum PTH levels (more than tenfold higher than control) and whose excised parathyroid glands were insensitive to $Ca_o^{2+}$ are reported. Breeding of parathyroid cell-specific CaSR KO mice and procedures for isolation and ex vivo culture of their parathyroid glands have been approved (Protocol 18-017) by the Institutional Animal Care and Use Committee (IACUC) at San Francisco Veterans Affairs Medical Center. We confirm that we have complied with all relevant ethical regulations for animal testing and research.

**Amino acid conservation analysis**. A data set of family C GPCR alignments was generated by Herberger et al.[29] and included 138 sequences, with 42 of them comprising the CaSR subgroup. Exploratory data analysis, data visualization, and database management was performed with R-studio by using the "seqinr" package[52,53]. Weblogo (version 3.0[54]) was used to visualize amino acid conservation in the four Pi-binding sites predicted by Geng et al.[27]. Sequence conservation by position was assessed by bit-score calculation for each amino acid residue position in the four Pi-binding sites, with the relative frequency of each amino acid displayed pictorially by height of letter.

**In silico study of CaSR-ECD anion-binding sites**. CaSR-ECD original structures were obtained from Protein Data Bank (PDB) by using the accession numbers 5K5S (active conformation) and 5K5T (inactive conformation). Anion-binding interactions were studied by using PyMol Molecular Graphics System, Schrodinger[55].

**Data analysis**. Changes in Fura2 ratio (340/380) were measured with MetaFluor imaging system and assessed by determining the area under the curve by using the R-studio "PKNCA Perform Pharmacokinetic Non-Compartmental Analysis" package[52,56]. Responses were measured for 190 s, but excluding the first 40–50 s to

avoid change artifacts. Densitometry of immunoreactive bands was performed by using Image Lab 6.0 2D software (BIO-RAD). GraphPad Prism 7 (GraphPad Software, San Diego, CA, USA) was used for nonlinear regression analysis, mean, and SEM. Orthosteric concentration−response curves were normalized to 100% as the maximum orthosteric agonist/antagonist response. Normalized data were fitted to a four-parameter concentration response for $Ca_i^{2+}$-mobilization with Hill Eq. (1), where Top and Bottom are the maximal and the minimal asymptotes of the concentration–response curve.

$$Y = (\text{Bottom}) + \frac{(\text{Top}) - (\text{Bottom})}{1 + \frac{\text{X}}{\text{EC}_{50}}^{(\text{Hill coefficient})}}. \qquad (1)$$

**Statistical analysis**. Statistical differences between the $EC_{50}$ and $E_{max}$ in the presence and absence of Pi were determined by an extra sum-of- squares F-test. A one-way ANOVA with Dunnett's multiple comparisons post test was used to determine statistical differences between $EC_{50}$ and $E_{max}$. A one-way repeated measure ANOVA, with Dunnett's for normally distributed data or Friedman Dunn's for non-normally distributed data, was used to determine statistical differences between 0, 0.8, and 2 mM Pi in $Ca_i^{2+}$-mobilization, PTH secretion, and pERK assays. Statistical differences between 0.8 and 2 mM Pi in pERK in WT and R62A were determined by Student's t test. Data are presented as mean ± SEM or median with range, except for $EC_{50}$ and $IC_{50}$ values, which are presented as mean 95% confidence interval, and P values <0.05 were considered statistically significant. Replicates are reported either as n for the number of coverslips used per experiment, or as N for the number of independent experiments or animals.

**Reporting summary**. Further information on research design is available in the Nature Research Reporting Summary linked to this article.

## Data availability
The source data underlying Figs. 1–4 and 7, and all the Supplementary Figs. are provided as a separate source Data file. The data sets generated and/or analyzed during the current study are available from the corresponding author upon reasonable request.

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

## Acknowledgements

This project was funded from the Marie Sklodowska-Curie Actions of the European Union's Horizon 2020 program under grant agreement no. 675228 as part of the CaSR Biomedicine Network. Travel grants from Boehringer Ingelheim Fonds and the British Pharmacological Society (Schachter Award) supported research visits to the laboratories of Prof. Chang (where the work was funded from NIH R01DK121656 and VA Merit Review I01BX003453) and Prof. Conigrave, respectively. We thank Prof. Emeritus Leigh Delbridge (Mater Misericordiae Hospital, North Sydney, NSW) and Prof. Stan Sidhu (Royal North Shore Hospital, St. Leonards, NSW) for providing the human parathyroid tissue used for PTH secretion experiments. In addition, we thank Anne Shenton (Clinical Biochemistry, Manchester Royal Infirmary) for measuring free calcium levels in the experimental solutions, as well as Rebecca Kirkwood-Wilson, Lukša Popović, Sarah Maguire, and Kimberly Edwards for their assistance in preliminary experiments and Prof. Jim Warwicker for discussion of the crystal structure. In memory of Dr Vasken Ohanian.

## Author contributions

P.P.C. designed and performed the experiments, provided bioinformatic analysis, and wrote and edited the paper. A.H. performed murine PTH secretion experiments with assistance from C.T. H.C.M. performed human PTH experiments. E.F.N. contributed to experimental design and provided useful discussion. W.C. provided the animal model, performed murine PTH experiments, and provided useful discussion. A.D.C. provided human parathyroid samples, assisted in human PTH secretion analysis, and provided useful discussion. D.T.W. conceived the project, designed experiments, and wrote and edited the paper.

## Competing interests

The authors declare no competing interests.
