## [Peer Review File · Nature Communications]

Reviewers' comments:

Reviewer #1 (Remarks to the Author):

Review NCOMMS-19-01486

The paper by Centeno PP et al on: " Phosphate acts directly on the calcium-sensing receptor to stimulate parathyroid hormone secretion" describes the results of an interesting investigation, based on a novel concept that not only calcium, but also inorganic plasma phosphate (Pi) is mediating its effect on parathyroid gland secretion of PTH via the calcium sensing receptor (CaSR). For years a relationship between Pi and PTH was known to exist, but not described in specific details before in 1996, where a direct effect of Pi on PTH secretion was shown by ref 25 and 26, and by Almaden Y et al (JBMR, 11:970-976; 1996), a ref that should be added. Until then the effect of Pi was considered probably to be mediated primarily via induced changes in the calcium homeostasis. The present investigation represents an important and interesting step forward in the understanding of the relationship between Pi and the parathyroids. The concept is original and the methods used relevant and up to date.

Comments:

It is surprising that the main focus of the present manuscript is on CKD-MBD and associated hyperphosphatemia. Throughout the manuscript the authors deal with the relevance of their concept for patients with chronic kidney failure. None of the in vivo and in vitro models used are CKD models and none used the secondary hyperparathyroidism in CKD and the authors have not examined the relationship between Pi and CaSR in parathyroid glands from uremic patients or experimental CKD models. Therefore, it is suggested that focus in the manuscript should be directed more on "normal" physiology and pathophysiology rather than on exclusively on CKD.

In this respect the focus of the present investigation is "only" on Pi levels within what is called pathophysiological concentrations and not on the "normal" physiological range or on hypophosphatemia. Do the authors believe that there is no relevance of phosphate sensing by CaSR in normal physiology or in conditions with low Pi?

Why start the Abstract with: "Chronic kidney disease mineral and bone disorder (CKD-MBD) leads to vascular calcification and cardiovascular mortality in advanced renal disease. CKD-MBD is typically associated with hyperphosphataemia and secondary hyperparathyroidism, which contribute to increased mortality in kidney dialysis patients.", when this is not what is studied in the present investigation. Similar considerations might relate to all the aspects of CKD-MBD in the introduction, which are not further examined.

Until now no available line of parathyroid cells exists, which can be used experimentally, as all attempts have resulted in a rapid loss of the ability to sense calcium and secrete PTH. Therefore, studying parathyroid secretion have to use freshly prepared parathyroid cell preparations, either obtained from humans, bovine or experimental animals.

Presence of a CaSR has been demonstrated in nearly all organs where it has been searched for, and the sensitivity of the CaSR for calcimimetics differs, when compared to the sensitivity for calcium in the parathyroids. This phenomenon may make the results from using non-parathyroid cells difficult in the interpretation of the response of parathyroids to a given stimulus. Furthermore, it's important also to discuss whether these many organs with CaSR also have responsiveness to phosphate.

One might get the impression from this manuscript that Pi is the most important stimulus for the development of hypersecretion of PTH in uremia. There are however besides calcium also many other important factors involved, e.g. calcitriol, VDR, FGF23, Klotho and intrinsic factors and the interrelation feed back mechanisms between these. It is suggested that this focus should be discussed a little further.

The main model in this investigation uses the calcimimetic, NPS R568, in order to activate the CaSR and at the same time keep the extracellular calcium conc low in order to avoid calcium and phosphate precipitation. It has previously been shown that such a precipitation takes hours and will probably not be relevant in the present models on the very rapid effects of changes in

extracellular calcium and phosphate. It would be interesting and relevant to see the results of experiments with different calcium conc without the use of R568.

The authors focus correctly early in their investigation on the fact that high Pi binds to calcium and that the demonstrated effects could be due to the decreased levels of calcium. Therefore, they have measured the free Ca²⁺ in the Pi-containing buffers (supplementary Fig 1) and found that the difference in Ca²⁺ was less than 5%. This result is crucial for the whole concept studied in the present investigation, but is only supported by very few data points and without any SD or SEM and no number of experiments performed. Please, elaborate, and also present the time frame of these experiments.

P.9: The authors found that in glands from the KOCasr mice in which CaSR had been functionally knocked-out, Pi -mediated stimulation of PTH secretion was not observed (Fig. 4c). They conclude: "These results are consistent with the previous human PTH secretion experiments, concluding that the Pi -induced rise in PTH secretion is mediated by the CaSR." However, the KOCasr had maximally stimulated PTH secretion and the Fig 4c indicates that when PTH is maximally stimulated as in normal mice with intact CaSR at a Ca²⁺ of 0.8 mM increasing Pi levels can not further stimulate PTH secretion. This needs to be discussed.

What was the diagnosis of the "normal subjects/patients", who needed neck surgery ? They can't have been normal! The diagnoses should be presented, and the plasma calcium and phosphate levels given, as a minimum.

The expression "Kidney dialysis patients" in the Abstract should be changed to " patients with chronic kidney disease on dialysis"

p.17, lines 262 -264. The sentence "Indeed, such a rapid increase in PTH secretion and in response to such small, pathophysiologic increases in Pi concentration has never been reported before." Is not absolutely correct. The authors are kindly referred to: Martin DR, Ritter CS, Slatopolsky E, Brown AJ: Acute regulation of parathyroid hormone by dietary phosphate. *Am J Physiol Endocrinol Metab.* 289:E729-34;2005 showing that intravenous infusion of sodium phosphate increased PTH within 10 min, whereas infusion of sodium chloride had no effect.

Dulbeccos' modified Eagle's medium was used for cell culture. It was added 10% FBS (containing calcium). What were the calcium and phosphate concentrations in the basal medium?

The human parathyroid cell preparations were kept in MEM for up to 48 hours. A problem is that cultured parathyroid cells rapidly lose their responsiveness to calcium due to decreased expression of the CaSR, as shown by Brown AJ et al in *Biochem Biophys Res Commun*, 212: 861-867; 1995 (should be discussed and the ref added). The authors only have parathyroid cell preparations from a few humans and therefore the response to calcium should be presented for each individual preparation as the response might differ significantly due to the time frame after PTX.

How was it ensured that the two murine parathyroid glands were "free of thyroid tissue (also containing CaSR) and connective tissue" ? It's a very difficult procedure in the mouse.

When looking at the references, then some references should be added as mentioned, but the references also need to be carefully revised, just e.g. ref 12: where "the official journal of the American Society for Bone and mineral Research" should be deleted. Just use the name of the journal only. The same is the case for a number of other references.

Furthermore some references, e.g. ref 13, lack the name of the journal, vol and pages, etc. Please, carefully check all the references.

Reviewer #2 (Remarks to the Author):

Centeno & colleagues have elucidated an important biological mechanism by which the parathyroid gland may detect alterations in Pi concentrations. Indeed, they have demonstrated that Pi sensing may actually be mediated by the parathyroid Ca-sensing receptor, and defined a specific extracellular domain residue, Arg62, which is involved in Pi binding.

They have undertaken a comprehensive & well thought out set of studies to evaluate the effect of extracellular Pi on CaSR function. Each experiment appears to have been carefully planned with due consideration given to confounding factors such as the effects of pH and potential alterations in ionised Ca. My only reservation is regarding the studies involving the CaSR ko mouse parathyroid glands (PTGs) as these glands are already likely to be maximally secreting PTH, and therefore Pi stimulation would not be expected to increase PTH secretion further. Perhaps the authors could consider addressing this point in the discussion section.

Overall, the manuscript is well written and the clinical relevance of the study findings are nicely outlined in the discussion section.

I have some additional minor comments, which are outlined below:

Figures - each panel should be individually labelled. So, if there are 9 panels, then these need to be labelled consecutively as a-i

Page 2, line 29: do increases in Ca x P really contribute to skeletal fracture?

Page 5, Fig 1: panel B is unlabelled

Page 8 Fig 3:

Please clarify what the term "% response is referring to"

How many independent patients were the human parathyroid samples collected from?

Page 9, line 149: Please cite refs for the previous human PTH secretion experiments

Page 10, Fig 4: 1.6 mM Ca appears to be lowering the PTH responses of CaSR ko mouse PTGs compared to CaSR ko PTGs treated with 1.2 mM Ca. Is this correct?

Page 12: The sentence that "These features highlight the importance of Ca²⁺ binding site 3 and Pi binding sites 1 and 3 as the key conformational events that govern CaSR's activity" is too strong, as there may be other conformational events which are of equal or greater importance for regulating receptor function

Page 15, Fig 7: Please clarify why the experiment was conducted with 1mM Ca & not 0.5 mM Ca

Reviewers' comments

Reviewer #1 (Remarks to the Author):

Review NCOMMS-19-01486

The paper by Centeno PP et al on: "Phosphate acts directly on the calcium-sensing receptor to stimulate parathyroid hormone secretion" describes the results of an interesting investigation, based on a novel concept that not only calcium, but also inorganic plasma phosphate (Pi) is mediating its effect on parathyroid gland secretion of PTH via the calcium sensing receptor (CaSR). For years a relationship between Pi and PTH was known to exist, but not described in specific details before in 1996, where a direct effect of Pi on PTH secretion was shown by ref 25 and 26, and by Almaden Y et al (JBMR, 11:970-976; 1996), a ref that should be added. Until then the effect of Pi was considered probably to be mediated primarily via induced changes in the calcium homeostasis. The present investigation represents an important and interesting step forward in the understanding of the relationship between Pi and the parathyroids. The concept is original and the methods used relevant and up to date.

Comments:

It is surprising that the main focus of the present manuscript is on CKD-MBD and associated hyperphosphatemia. Throughout the manuscript the authors deal with the relevance of their concept for patients with chronic kidney failure. None of the in vivo and in vitro models used are CKD models and none used the secondary hyperparathyroidism in CKD and the authors have not examined the relationship between Pi and CaSR in parathyroid glands from uremic patients or experimental CKD models. Therefore, it is suggested that focus in the manuscript should be directed more on "normal" physiology and pathophysiology rather than on exclusively on CKD.

In this respect the focus of the present investigation is "only" on Pi levels within what is called pathophysiological concentrations and not on the "normal" physiological range or on hypophosphatemia. Do the authors believe that there is no relevance of phosphate sensing by CaSR in normal physiology or in conditions with low Pi?

Why start the Abstract with: "Chronic kidney disease mineral and bone disorder (CKD-MBD) leads to vascular calcification and cardiovascular mortality in advanced renal disease. CKD-MBD is typically associated with hyperphosphataemia and secondary hyperparathyroidism, which contribute to increased mortality in kidney dialysis patients.", when this is not what is studied in the present investigation. Similar considerations might relate to all the aspects of CKD-MBD in the introduction, which are not further examined.

Until now no available line of parathyroid cells exists, which can be used experimentally, as all attempts have resulted in a rapid loss of the ability to sense calcium and secrete PTH. Therefore, studying parathyroid secretion have to use freshly prepared parathyroid cell preparations, either obtained from humans, bovine or experimental animals.

Presence of a CaSR has been demonstrated in nearly all organs where it has been searched for, and

the sensitivity of the CaSR for calcimimetics differs, when compared to the sensitivity for calcium in the parathyroids. This phenomenon may make the results from using non-parathyroid cells difficult in the interpretation of the response of parathyroids to a given stimulus. Furthermore, it's important also to discuss whether these many organs with CaSR also have responsiveness to phosphate.

One might get the impression from this manuscript that Pi is the most important stimulus for the development of hypersecretion of PTH in uremia. There are however besides calcium also many other important factors involved, e.g. calcitriol, VDR, FGF23, Klotho and intrinsic factors and the interrelation feed back mechanisms between these. It is suggested that this focus should be discussed a little further.

The main model in this investigation uses the calcimimetic, NPS R568, in order to activate the CaSR and at the same time keep the extracellular calcium conc low in order to avoid calcium and phosphate precipitation. It has previously been shown that such a precipitation takes hours and will probably not be relevant in the present models on the very rapid effects of changes in extracellular calcium and phosphate. It would be interesting and relevant to see the results of experiments with different calcium conc without the use of R568.

The authors focus correctly early in their investigation on the fact that high Pi binds to calcium and that the demonstrated effects could be due to the decreased levels of calcium. Therefore, they have measured the free Ca²⁺ in the Pi-containing buffers (supplementary Fig 1) and found that the difference in Ca²⁺ was less than 5%. This result is crucial for the whole concept studied in the present investigation, but is only supported by very few data points and without any SD or SEM and no number of experiments performed. Please, elaborate, and also present the time frame of these experiments.

P.9: The authors found that in glands from the KO_{CaSR} mice in which CaSR had been functionally knocked-out, Pi -mediated stimulation of PTH secretion was not observed (Fig. 4c). They conclude: "These results are consistent with the previous human PTH secretion experiments, concluding that the Pi -induced rise in PTH secretion is mediated by the CaSR." However, the KO_{CaSR} had maximally stimulated PTH secretion and the Fig 4c indicates that when PTH is maximally stimulated as in normal mice with intact CaSR at a Ca²⁺ of 0.8 mM increasing Pi levels can not further stimulate PTH secretion. This needs to be discussed.

What was the diagnosis of the "normal subjects/patients", who needed neck surgery ? They can't have been normal! The diagnoses should be presented, and the plasma calcium and phosphate levels given, as a minimum.

The expression "Kidney dialysis patients" in the Abstract should be changed to " patients with chronic kidney disease on dialysis"

p.17, lines 262 -264. The sentence "Indeed, such a rapid increase in PTH secretion and in response to such small, pathophysiologic increases in Pi concentration has never been reported before." Is not absolutely correct. The authors are kindly referred to: Martin DR, Ritter CS, Slatopolsky E, Brown AJ:

Acute regulation of parathyroid hormone by dietary phosphate. Am J Physiol Endocrinol Metab. 289:E729-34;2005 showing that intravenous infusion of sodium phosphate increased PTH within 10 min, whereas infusion of sodium chloride had no effect.

Dulbeccos' modified Eagle's medium was used for cell culture. It was added 10% FBS (containing calcium). What were the calcium and phosphate concentrations in the basal medium?

The human parathyroid cell preparations were kept in MEM for up to 48 hours. A problem is that cultured parathyroid cells rapidly lose their responsiveness to calcium due to decreased expression of the CaSR, as shown by Brown AJ et al in Biochem Biophys Res Commun, 212: 861-867; 1995 (should be discussed and the ref added). The authors only have parathyroid cell preparations from a few humans and therefore the response to calcium should be presented for each individual preparation as the response might differ significantly due to the time frame after PTX.

How was it ensured that the two murine parathyroid glands were "free of thyroid tissue (also containing CaSR) and connective tissue"? It's a very difficult procedure in the mouse.

When looking at the references, then some references should be added as mentioned, but the references also need to be carefully revised, just e.g. ref 12: where "the official journal of the American Society for Bone and mineral Research" should be deleted. Just use the name of the journal only. The same is the case for a number of other references.

Furthermore some references, e.g. ref 13, lack the name of the journal, vol and pages, etc. Please, carefully check all the references.

Reviewer #2 (Remarks to the Author):

Centeno & colleagues have elucidated an important biological mechanism by which the parathyroid gland may detect alterations in Pi concentrations. Indeed, they have demonstrated that Pi sensing may actually be mediated by the parathyroid Ca-sensing receptor, and defined a specific extracellular domain residue, Arg62, which is involved in Pi binding.

They have undertaken a comprehensive & well thought out set of studies to evaluate the effect of extracellular Pi on CaSR function. Each experiment appears to have been carefully planned with due consideration given to confounding factors such as the effects of pH and potential alterations in ionised Ca. My only reservation is regarding the studies involving the CaSR ko mouse parathyroid glands (PTGs) as these glands are already likely to be maximally secreting PTH, and therefore Pi stimulation would not be expected to increase PTH secretion further. Perhaps the authors could consider addressing this point in the discussion section.

Overall, the manuscript is well written and the clinical relevance of the study findings are nicely outlined in the discussion section.

I have some additional minor comments, which are outlined below:

Figures - each panel should be individually labelled. So, if there are 9 panels, then these need to be

labelled consecutively as a-l Page 2, line 29: do increases in Ca x P really contribute to skeletal fracture?

Page 5, Fig 1: panel B is unlabelled Page 8 Fig 3: Please clarify what the term "% response is referring to" How many independent patients were the human parathyroid samples collected from? Page 9, line 149: Please cite refs for the previous human PTH secretion experiments Page 10, Fig 4: 1.6 mM Ca appears to be lowering the PTH responses of CaSR ko mouse PTGs compared to CaSR ko PTGs treated with 1.2 mM Ca. Is this correct?

Page 12: The sentence that "These features highlight the importance of Ca²⁺ binding site 3 and Pi binding sites 1 and 3 as the key conformational events that govern CaSR's activity" is too strong, as there may be other conformational events which are of equal or greater importance for regulating receptor function

Page 15, Fig 7: Please clarify why the experiment was conducted with 1mM Ca & not 0.5 mM Ca

Response to reviewers

Reviewer #1 (Remarks to the Author):

The paper by Centeno PP et al on: "Phosphate acts directly on the calcium-sensing receptor to stimulate parathyroid hormone secretion" describes the results of an interesting investigation, based on a novel concept that not only calcium, but also inorganic plasma phosphate (Pi) is mediating its effect on parathyroid gland secretion of PTH via the calcium sensing receptor (CaSR). For years a relationship between Pi and PTH was known to exist, but not described in specific details before in 1996, where a direct effect of Pi on PTH secretion was shown by ref 25 and 26, and by Almaden Y et al (JBMR, 11:970-976; 1996), a ref that should be added. Until then the effect of Pi was considered probably to be mediated primarily via induced changes in the calcium homeostasis. The present investigation represents an important and interesting step forward in the understanding of the relationship between Pi and the parathyroids. The concept is original and the methods used relevant and up to date.

Response: We thank Reviewer #1 for the encouragement and advice. We have added the suggested reference in the revised manuscript (reference 5).

Comments:

1. It is surprising that the main focus of the present manuscript is on CKD-MBD and associated hyperphosphatemia. Throughout the manuscript the authors deal with the relevance of their concept for patients with chronic kidney failure. None of the in vivo and in vitro models used are CKD models and none used the secondary hyperparathyroidism in CKD and the authors have not examined the relationship between Pi and CaSR in parathyroid glands from uremic patients or experimental CKD models. Therefore, it is suggested that focus in the manuscript should be directed more on "normal" physiology and pathophysiology rather than on exclusively on CKD.

In this respect the focus of the present investigation is “only” on Pi levels within what is called pathophysiological concentrations and not on the “normal” physiological range or on hypophosphatemia.

Response: We agree with this comment and have rewritten key sections of the manuscript to focus much more on physiological phosphate homeostasis (0.8-1.5 mM) and limited our discussion of CKD-MBD (>1.5 mM).

2. Do the authors believe that there is no relevance of phosphate sensing by CaSR in normal physiology or in conditions with low Pi?

Response: We thank the reviewer for pointing this out. Indeed we do believe that phosphate-sensing by the CaSR may be highly relevant in normal physiology. Specifically, CaSR phosphate sensitivity was found to be greater in the physiological range (0.8-1.4 mM) than in the CKD range (>1.5 mM). Thus by increasing PTH secretion via CaSR inhibition, phosphate would be able to enhance its renal excretion to avoid hyperphosphatemia.

Regarding “low Pi”, we did not observe any gross changes in wild-type CaSR activity between 0 and 0.8 mM Pi but did not specifically test any concentrations equivalent to pathophysiological hyposphosphatemia. This would be interesting to investigate in future work. As a result, changes have been made to the text in Results and Discussion.

3. Why start the Abstract with: “Chronic kidney disease mineral and bone disorder (CKD-MBD) leads to vascular calcification and cardiovascular mortality in advanced renal disease. CKD-MBD is typically associated with hyperphosphatemia and secondary hyperparathyroidism, which contribute to increased mortality in kidney dialysis patients.”, when this is not what is studied in the present investigation. Similar considerations might relate to all the aspects of CKD-MBD in the introduction, which are not further examined.

Response: We agree with this suggestion. Therefore, we have rewritten the Abstract to place the emphasis on physiological phosphate homeostasis, with the reference to CKD-MBD coming second. Similarly, the Introduction now starts with physiological Pi homeostasis, with the CKD-MBD connection deemphasised.

4. Until now no available line of parathyroid cells exists, which can be used experimentally, as all attempts have resulted in a rapid loss of the ability to sense calcium and secrete PTH. Therefore, studying parathyroid secretion have to use freshly prepared parathyroid cell preparations, either obtained from humans, bovine or experimental animals.

Response: We agree with Reviewer #1. The lack of a functional parathyroid cell line and the tendency of the CaSR to downregulate in culture are perennial limitations in the field. The rapid loss of CaSR expression has been observed in disaggregated parathyroid cells. However, we had access to freshly isolated human parathyroid undigested tissue that we were able to use usually within 24 hours (maximum 48 hours). To limit CaSR downregulation, the human tissue samples were stored at 4°C as sliced tissues cubes in

Hank's buffer until tissue disaggregation and cell preparation prior to immediate PTH secretion assay. The protocol reported (Figure 3) has been employed successfully for measuring CaSR-modulated human PTH secretion in a number of studies, in which CaSR expression remained functionally sufficient at least in the short-term [Conigrave *et al.* 2004 (J Biol Chem 279, 38151-9), Mun *et al.*, 2009 (J Clin Endocrinol Metab, 94:3567-74), Broadhead *et al.* 2011 (J Biol Chem 286:8786-97), McCormick *et al.* 2010 (J Biol Chem 85(19):14170-7), Campion *et al.* 2015 (J Am Soc Nephrol 26(9):2163-71)]. Regarding the murine parathyroid glands, PTH secretion assays were performed immediately after the tissue collection and this protocol has also been used successfully to measure CaSR-modulated PTH secretion [Chang *et al.* 2008 (Sci. Signal 1(35):ra1), Cheng *et al.* 2013 (J. Bone Miner. Res 28(5):1087-100)].

5. Presence of a CaSR has been demonstrated in nearly all organs where it has been searched for, and the sensitivity of the CaSR for calcimimetics differs, when compared to the sensitivity for calcium in the parathyroids. This phenomenon may make the results from using non-parathyroid cells difficult in the interpretation of the response of parathyroids to a given stimulus.

Response: We thank the Referee for this advice. We have thus addressed this specifically with the following change: "these values were obtained using CaSR-HEK cells whereas CaSR sensitivity for Ca²⁺, R568 and Pi differ in parathyroid cells".

6. Furthermore, it's important also to discuss whether these many organs with CaSR also have responsiveness to phosphate.

Response: We agree and have modified the text in Results and Discussion sections to include this point.

7. One might get the impression from this manuscript that Pi is the most important stimulus for the development of hypersecretion of PTH in uremia. There are however besides calcium also many other important factors involved, e.g. calcitriol, VDR, FGF23, Klotho and intrinsic factors and the interrelation feed back mechanisms between these. It is suggested that this focus should be discussed a little further.

Response: We accept this criticism and to avoid ambiguity have modified both the Introduction and Discussion to highlight the relative contributions of FGF23, vitamin D receptor and Klotho.

8. The main model in this investigation uses the calcimimetic, NPS R568, in order to activate the CaSR and at the same time keep the extracellular calcium conc low in order to avoid calcium and phosphate precipitation. It has previously been shown that such a precipitation takes hours and will probably not be relevant in the present models on the very rapid effects of changes in extracellular calcium and phosphate. It would be interesting and relevant to see the results of experiments with different calcium conc without the use of R568.

Response: This is a very interesting suggestion and as a result we have performed additional experiments with the new data shown in Figure 1b and Figure 7a. Specifically, we tested the effect of 2 mM Pi on CaSR-mediated Ca^{2+} mobilisation following stimulation with different concentrations of Ca^{2+} (2 and 4 mM) on CaSR^{WT} and on CaSR^{R62A} in the absence of any additional PAM or calcimimetic. As expected, Ca^{2+} -stimulated CaSR^{WT} activity was significantly reduced in the presence of 2 mM Pi whereas Ca^{2+} -stimulated CaSR^{R62A} activity was not altered. It should also be noted that other CaSR activators were used to induce CaSR activation, namely Cinacalcet (Figure 1e and 2c) and Spermine (Figure 1 c) with the same experimental outcome as when using R568.

9. The authors focus correctly early in their investigation on the fact that high Pi binds to calcium and that the demonstrated effects could be due to the decreased levels of calcium. Therefore, they have measured the free Ca^{2+} in the Pi-containing buffers (supplementary Fig 1) and found that the difference in Ca^{2+} was less than 5%. This result is crucial for the whole concept studied in the present investigation, but is only supported by very few data points and without any SD or SEM and no number of experiments performed. Please, elaborate, and also present the time frame of these experiments.

Response: We accept this point and thus have performed additional measurements of ionised Ca^{2+} concentration in these Pi-containing buffers. Supplementary Figure 1 now shows the resulting median ionised Ca^{2+} concentrations for each (3-5 replicates, including range).

10. P.9: The authors found that in glands from the KO*Casr* mice in which CaSR had been functionally knocked-out, Pi-mediated stimulation of PTH secretion was not observed (Fig. 4c). They conclude: "These results are consistent with the previous human PTH secretion experiments, concluding that the Pi-induced rise in PTH secretion is mediated by the CaSR." However, the KO*Casr* had maximally stimulated PTH secretion and the Fig 4c indicates that when PTH is maximally stimulated as in normal mice with intact CaSR at a Ca^{2+} of 0.8 mM increasing Pi levels can not further stimulate PTH secretion. This needs to be discussed.

Response: We agree and thank the reviewer for the insight and advice. We were aware of this limitation, however, it is currently not possible to limit PTH secretion in homozygous KO*Casr* mice. For example, heterozygous CaSR knockouts exhibit more moderately elevated PTH secretion but still retain their Pi sensitivity, while calcimimetics cannot be used to ameliorate the PTH secretion in KO*Casr* parathyroid glands as they act via the CaSR itself. It is not certain whether the high PTH secretion observed in the KO*Casr* parathyroid glands represents the maximal secretory capacity of the glands or not. Although not fully addressing this issue, we would point out that underactivation of the CaSR (using either low calcium concentration, or a calcilytic, or, with *Casr* containing a loss-of-function mutation) results in increased PTH mRNA expression. Thus, KO*Casr* parathyroid glands might conceivably have an increased PTH secretory capacity that would permit Pi to induce additional secretion in a CaSR-independent manner should such a mechanism exist. By generating a mouse model that expresses a CaSR that lacks the Pi-binding site, we intend to address this issue more fully in the future.

To address this limitation the following sentence has been added in the discussion: "It should be noted that parathyroid glands from *KOCasr* mice exhibit elevated PTH secretion and therefore the potential for further substantial increases in secretion may be impaired".

11. What was the diagnosis of the "normal subjects/patients", who needed neck surgery ? They can't have been normal! The diagnoses should be presented, and the plasma calcium and phosphate levels given, as a minimum.

Response: These are patients who have undergone total thyroidectomy for major thyroid disease (e.g., a large multinodular goitre or Grave's disease) in which the surgeon saved a parathyroid gland, minced it into pieces of around 1 mm³, and then returned the tissue into the ipsilateral sternomastoid muscle via a syringe. This has been shown to reduce post-operative hypoparathyroidism [Zedenious *et al* 1999 (Aust N Z J Surg. 69:794-7), Shaha *et al* 1991 (J Surg Oncol. 46:21-4), Shaha *et al* 1998 (Am J Otolaryngol. 19:113-7)]. Tissue fragments left in the syringe were provided to us (with the signed consent of the patients) but without identification. However, the underlying conditions were specifically thyroidal in nature and the parathyroid tissue was "healthy" i.e. not adenomatous. We rely on the surgeon to tell us if there was an unexpected non-thyroid diagnosis or a disturbance of the plasma calcium or phosphate levels. We understand that the plasma calcium and phosphate levels were normal in all of the patients from whom tissue samples were obtained, but we have no access to the clinical biochemistry values. In addition, the tissues were stored in ice-cold Hanks' balanced salt solution for at least 12 hours prior to cell isolation, permitting equilibration.

We have added further details about this procedure to the Methods section and also commented on the limitations of the technique in Results and Discussion.

12. The expression "Kidney dialysis patients" in the Abstract should be changed to " patients with chronic kidney disease on dialysis"

Due to other changes in the Abstract, this sentence has been removed.

13. p.17, lines 262 -264. The sentence "Indeed, such a rapid increase in PTH secretion and in response to such small, pathophysiologic increases in Pi concentration has never been reported before." Is not absolutely correct. The authors are kindly referred to: Martin DR, Ritter CS, Slatopolsky E, Brown AJ: Acute regulation of parathyroid hormone by dietary phosphate. Am J Physiol Endocrinol Metab. 289:E729-34;2005 showing that intravenous infusion of sodium phosphate increased PTH within 10 min, whereas infusion of sodium chloride had no effect.

Response: We agree and thus have added this reference and modified the Discussion text accordingly:

"These results are consistent with a previous *in vivo* study performed in rats where *Pi* injections into the duodenum or intravenous stimulated rapid (within 10 minutes) increases in plasma PTH, and the authors hypothesised the existence of a cell surface "phosphate

sensor", similar to the Ca^{2+} sensor, and expressed in the duodenum and the parathyroid gland (Martin 2005)."

14. Dulbeccos' modified Eagle's medium was used for cell culture. It was added 10% FBS (containing calcium). What were the calcium and phosphate concentrations in the basal medium?

Response: Cell culture media used for CaSR-HEK cells contained 1.8 mM calcium and 0.9 mM phosphate, both values are close to human physiological levels. Details have been added in the methods section as follows: "...cells were grown in Dulbeccos' modified Eagle's medium containing 1.8 mM $CaCl_2$ and 0.9 mM Na_2HPO_4 from Sigma (D5796)..."

15. The human parathyroid cell preparations were kept in MEM for up to 48 hours. A problem is that cultured parathyroid cells rapidly lose their responsiveness to calcium due to decreased expression of the CaSR, as shown by Brown AJ et al in *Biochem Biophys Res Commun*, 212: 861-867; 1995 (should be discussed and the ref added). The authors only have parathyroid cell preparations from a few humans and therefore the response to calcium should be presented for each individual preparation as the response might differ significantly due to the time frame after PTX.

Response: We agree with the reviewer and are aware of the time-dependent decline in CaSR expression following tissue disaggregation, as discussed above (see Comment 4). We would clarify that human tissue samples were stored as sliced tissue cubes (undigested tissue). Tissue disaggregation and cell preparation were conducted immediately before PTH secretion assay and within 48 hours at the most (the majority were begun with 24 hours). In addition and as requested, we have added a Supplementary Figure containing the PTH secretion trace from every individual experiment (Supplementary Fig. 4).

The suggested reference has been added to the Discussion with a statement clarifying the experimental conditions.

16. How was it ensured that the two murine parathyroid glands were "free of thyroid tissue (also containing CaSR) and connective tissue" ? It's a very difficult procedure in the mouse.

Response: We agree that the dissection of murine parathyroid glands is a particularly difficult procedure and requires significant skill and experience. However, Professor Wenhan Chang is the world's leading murine parathyroid expert and has published several studies employing the same procedure [Cheng *et al* 2013 (*J Bone Miner Res*. 28(5):1087-100), Chang *et al* 2008 (*Sci Signal* 1(35):ra1)]. These references have now been added to the manuscript.

As to whether the parathyroid glands may have thyroid cells attached, Drs Chang, Ward and Conigrave have performed transcriptomic microarray analyses (unpublished) on mouse parathyroid glands. Four separate murine gene microarrays (Affymetrix) were performed using RNA collected from separate batches of mouse parathyroid glands (~20 glands from 10 mice in each batch) and all dissected by Prof. Chang. Significant gene expression of PTH, Vitamin D receptor and CaSR was observed, whereas the expression of thyroid stimulating hormone receptor (TSHR) was determined to be "Absent" in all 4 of the arrays which each

contained three different TSHR probe sets. Therefore, the authors are confident that the glands do not contain detectable levels of thyroid cells.

17. When looking at the references, then some references should be added as mentioned, but the references also need to be carefully revised, just e.g. ref 12: where "the official journal of the American Society for Bone and mineral Research" should be deleted. Just use the name of the journal only. The same is the case for a number of other references. Furthermore some references, e.g. ref 13, lack the name of the journal, vol and pages, etc. Please, carefully check all the references.

Response: We have carefully reviewed the reference list and believe that all are now complete and in conformity with the journal style.

Reviewer #2 (Remarks to the Author):

Centeno & colleagues have elucidated an important biological mechanism by which the parathyroid gland may detect alterations in Pi concentrations. Indeed, they have demonstrated that Pi sensing may actually be mediated by the parathyroid Ca-sensing receptor, and defined a specific extracellular domain residue, Arg62, which is involved in Pi binding.

They have undertaken a comprehensive & well thought out set of studies to evaluate the effect of extracellular Pi on CaSR function. Each experiment appears to have been carefully planned with due consideration given to confounding factors such as the effects of pH and potential alterations in ionised Ca. My only reservation is regarding the studies involving the CaSR ko mouse parathyroid glands (PTGs) as these glands are already likely to be maximally secreting PTH, and therefore Pi stimulation would not be expected to increase PTH secretion further. Perhaps the authors could consider addressing this point in the discussion section.

Overall, the manuscript is well written and the clinical relevance of the study findings are nicely outlined in the discussion section.

Response: We thank the reviewer for the advice. Regarding the studies involving the *KOCaSR* mouse parathyroid glands, we agree and thank the reviewer for this insightful comment (see also Comment 10 of Reviewer 1 above).

We were aware of this limitation; however, it is currently not possible to limit PTH secretion in homozygous *KOCaSR* mice. For example, heterozygous CaSR knockouts exhibit more moderately elevated PTH secretion but still retain their Pi sensitivity, while calcimimetics cannot be used to ameliorate the PTH secretion in *KOCaSR* parathyroid glands as they act via the CaSR itself. It is not certain whether the high PTH secretion observed in the *KOCaSR* parathyroid glands represents the maximal secretory capacity of the glands or not. Although not fully addressing this issue, we would point out that underactivation of the CaSR (using either low calcium concentration, or a calcilytic, or, with CaSRs containing a loss-of-function mutation) results in increased PTH mRNA expression. Thus, *KOCaSR* parathyroid glands might conceivably have an increased PTH secretory capacity that would permit Pi to induce additional secretion in a CaSR-independent manner should such a mechanism exist. By

generating a mouse model that expresses a CaSR that lacks the Pi-binding site we intend to address this issue more fully in the future.

This limitation has been addressed in the Discussion.

Minor comments

Figures - each panel should be individually labelled. So, if there are 9 panels, then these need to be labelled consecutively as a-i

Response: We apologise for the ambiguity caused. Journal policy states that "Each panel is labelled with a single letter, panels are not subdivided". However, we have made small adjustments to improve Figure clarity while remaining in accordance with the manuscript checklist.

Page 2, line 29: do increases in Ca x P really contribute to skeletal fracture?

Response: Thank you for pointing this issue out. Our wording did not represent what we had intended and thus "skeletal fracture" has been removed from the sentence.

Page 5, Fig 1: panel B is unlabelled

Response: Thank you for pointing this out. The panel has now been labelled.

Page 8 Fig 3: Please clarify what the term "% response is referring to"

Response: The Y-axis title has been changed to "PTH secretion (% Baseline)", with this defined in the legend as "Data normalised to baseline (initial exposure to 1.2mM Ca²⁺ / 0.8 mM Pi)".

How many independent patients were the human parathyroid samples collected from?

Response: All the human parathyroid samples used belong to independent patients. Thus, the text has been modified to "Data from N=9 samples from independent patients." In addition, a supplementary figure containing the individual traces from the parathyroid preparations from each patient, which includes the response to Ca²⁺ and Pi has been included (Supplementary Fig. 4).

Page 9, line 149: Please cite refs for the previous human PTH secretion experiments.

Response: We apologise for the ambiguity. Our sentence was intended to refer to the earlier (human parathyroid) experiment in the current study (Figure 3) and not to a previously published paper. Therefore, the sentence causing this confusion has been removed.

Page 10, Fig 4: 1.6 mM Ca appears to be lowering the PTH responses of CaSR ko mouse PTGs compared to CaSR ko PTGs treated with 1.2 mM Ca. Is this correct?

Response: It appears to be a very small fluctuation in KOCaSR PTH secretion, which occurs when changing calcium concentration. However, this marginal change is not significant and appears to result from biological variability.

Page 12: The sentence that "These features highlight the importance of Ca²⁺ binding site 3 and Pi binding sites 1 and 3 as the key conformational events that govern CaSR's activity" is too strong, as there may be other conformational events which are of equal or greater importance for regulating receptor function

Response: We agree and thus this sentence has been changed as follows: "These features highlight the importance of Ca²⁺ binding site 3 and Pi binding sites 1 and 3 at regulating CaSR's activity". In addition, in order to facilitate the reader's comprehension we have uploaded a Movie (Supplementary Movie 1) showing an animation of the CaSR's structure from its active (closed) to inactive (open) conformation, highlighting the R62-E277 salt bridge (pink sticks), R66-S301 hydrogen bond (green sticks), and Pi binding.

Page 15, Fig 7: Please clarify why the experiment was conducted with 1mM Ca & not 0.5 mM Ca

Response: Thank-you for pointing this issue out. The CaSR^{R62A} mutant exhibits a moderate reduction in Ca²⁺_o sensitivity (Supplementary Figure 6; EC₅₀=5.2 mM for R62A vs EC₅₀=3.4 mM for WT). Therefore, higher concentrations of Ca²⁺ were used (in the presence of R568) to achieve equivalent stimuli. This explanation has been included in the text.

REVIEWERS' COMMENTS:

Reviewer #1 (Remarks to the Author):

The authors have responded to all my previous comments and questions in a satisfactory way. Additional supporting experiments have been performed and the manuscript is clearly improved.

I have no more questions or comments.

Very interesting study

Congratulations

Klaus Olgaard

Reviewer #2 (Remarks to the Author):

The authors have satisfactorily addressed my queries. I have no further comments.

Dr Fadil Hannan

REVIEWERS' COMMENTS:

Reviewer #1 (Remarks to the Author):

The authors have responded to all my previous comments and questions in a satisfactory way. Additional supporting experiments have been performed and the manuscript is clearly improved. I have no more questions or comments.
Very interesting study
Congratulations

Klaus Olgaard

Reviewer #2 (Remarks to the Author):

The authors have satisfactorily addressed my queries. I have no further comments.

Dr Fadil Hannan

AUTHORS' RESPONSE

The authors would like to thank Dr Klaus Olgaard and Dr Fadil Hannan for their insightful comments and guidance, which we believe have improved the manuscript.